# FourierGNN: Rethinking Multivariate Time Series Forecasting from a Pure Graph Perspective

**Kun Yi**[1], **Qi Zhang**[2], **Wei Fan**[3], **Hui He**[1], **Liang Hu**[2], **Pengyang Wang**[4]
**Ning An**[5], **Longbing Cao**[6], **Zhendong Niu**[1]*
[1]Beijing Institute of Technology, [2]Tongji University, [3]University of Oxford
[4]University of Macau, [5]HeFei University of Technology, [6]Macquarie University
{yikun, hehui617, zniu}@bit.edu.cn, zhangqi_cs@tongji.edu.cn, weifan.oxford@gmail.com
railmilk@gmail.com, pywang@um.edu.mo, ning.g.an@acm.org, longbing.cao@mq.edu.au

## Abstract

Multivariate time series (MTS) forecasting has shown great importance in numerous industries. Current state-of-the-art graph neural network (GNN)-based forecasting methods usually require both graph networks (e.g., GCN) and temporal networks (e.g., LSTM) to capture inter-series (spatial) dynamics and intra-series (temporal) dependencies, respectively. However, the uncertain compatibility of the two networks puts an extra burden on handcrafted model designs. Moreover, the separate spatial and temporal modeling naturally violates the unified spatiotemporal inter-dependencies in real world, which largely hinders the forecasting performance. To overcome these problems, we explore an interesting direction of *directly applying graph networks* and rethink MTS forecasting from a *pure* graph perspective. We first define a novel data structure, *hypervariate graph*, which regards each series value (regardless of variates or timestamps) as a graph node, and represents sliding windows as space-time fully-connected graphs. This perspective considers spatiotemporal dynamics unitedly and reformulates classic MTS forecasting into the predictions on hypervariate graphs. Then, we propose a novel architecture *Fourier Graph Neural Network* (FourierGNN) by stacking our proposed Fourier Graph Operator (FGO) to perform matrix multiplications in *Fourier space*. FourierGNN accommodates adequate expressiveness and achieves much lower complexity, which can effectively and efficiently accomplish the forecasting. Besides, our theoretical analysis reveals FGO's equivalence to graph convolutions in the time domain, which further verifies the validity of FourierGNN. Extensive experiments on seven datasets have demonstrated our superior performance with higher efficiency and fewer parameters compared with state-of-the-art methods. Code is available at this repository: `https://github.com/aikunyi/FourierGNN`.

## 1 Introduction

Multivariate time series (MTS) forecasting plays an important role in numerous real-world scenarios, such as traffic flow prediction in transportation systems [1, 2], temperature estimation in weather forecasting [3, 4], and electricity consumption planning in the energy market [5, 6], etc. In MTS forecasting, the core challenge is to model intra-series (temporal) dependencies and simultaneously capture inter-series (spatial) correlations. Existing literature has primarily focused on the temporal modeling and proposed several forecasting architectures, including Recurrent Neural Network (RNN)-based methods (e.g., DeepAR [7]), Convolution Neural Network (CNN)-based methods (e.g., Temporal Convolution Network [8]) and more recent Transformer-based methods (e.g., Informer [9]

---

*Corresponding author

37th Conference on Neural Information Processing Systems (NeurIPS 2023).

and Autoformer [4]). In addition, another branch of MTS forecasting methods has been developed to not only model temporal dependencies but also places emphasis on spatial correlations. The most representative methods are the emerging Graph Neural Network (GNN)-based approaches [10, 2, 11] that have achieved state-of-the-art performance in the MTS forecasting task.

Previous GNN-based forecasting methods (e.g., STGCN [12] and TAMP-S2GCNets [11]) heavily rely on a pre-defined graph structure to specify the spatial correlations, which as a matter of fact cannot capture the *spatial dynamics*, i.e., the time-evolving spatial correlation patterns. Later advanced approaches (e.g., StemGNN [10], MTGNN [13], AGCRN [2]) can automatically learn inter-series correlations and accordingly model spatial dynamics without pre-defined priors, but almost all of them are designed by stacking graph networks (e.g., GCN and GAT) to capture *spatial dynamics* and temporal networks (e.g., LSTM and GRU) to capture *temporal dependencies*. However, the uncertain compatibility of the graph networks and the temporal networks puts extra burden on handcrafted model designs, which hinders the forecasting performance. Moreover, the respective modeling for the two networks *separately* learn spatial/temporal correlations, which naturally violate the real-world unified spatiotemporal inter-dependencies. In this paper, we explore an opposite direction of *directly applying graph networks for forecasting* and investigate an interesting question: ***can pure graph networks capture spatial dynamics and temporal dependencies even without temporal networks?***

To answer this question, we rethink the MTS forecasting task from a pure graph perspective. We start with building a new data structure, *hypervariate graph*, to represent time series with a united view of spatial/temporal dynamics. The core idea of the hypervariate graph is to construct a space-time fully-connected structure. Specifically, given a multivariate time series window (say input window) $X_t \in \mathbb{R}^{N \times T}$ at timestamp $t$, where $N$ is the number of series (variates) and $T$ is the length of input window, we construct a corresponding *hypervariate graph* structure represented as $\mathcal{G}_t^T = (X_t^T, A_t^T)$, which is initialized as a fully-connected graph of $NT$ nodes with adjacency matrix $A_t^T \in \mathbb{R}^{NT \times NT}$ and node features $X_t^T \in \mathbb{R}^{NT \times 1}$ by regarding *each value* $x_t^{(n)} \in \mathbb{R}^1$ (variate $n$ at step $t$) of input window as a distinct *node* of a hypervariate graph. Such a special structure design formulates both intra- and inter-series correlations of multivariate series as pure *node-node dependencies* in the hypervariate graph. Different from classic formulations that make spatial-correlated graphs and learn dynamics in a two-stage (spatial and temporal) process [13], our perspective views spatiotemporal correlations as a whole. It abandons the uncertain compatibility of spatial/temporal modeling, constructs adaptive space-time inter-dependencies, and brings up higher-resolution fusion across multiple variates and timestamps in MTS forecasting.

Then, with such a graph structure, the multivariate forecasting can be originally formulated into the predictions on the hypervariate graph. However, the node number of the hypervariate graph increase with the number of series ($N$) and the window length ($T$), leading to a graph of large order and size. This could make classic graph networks (e.g., GCN [14], GAT [15]) computationally expensive (usually with quadratic complexity) and suffer from optimization difficulty in obtaining accurate node representations [16]. To this end, we propose a novel architecture, *Fourier Graph Neural Network* (FourierGNN), for MTS forecasting from a pure graph perspective. Specifically, FourierGNN is built upon our proposed *Fourier Graph Operator* (FGO), which as a replacement of classic graph operation units (e.g., convolutions), performs *matrix multiplications* in *Fourier space* of graphs. By stacking FGO layers in Fourier space, FourierGNN can accommodate adequate learning expressiveness and in the mean time achieve much lower complexity (Log-linear complexity), which thus can effectively and efficiently accomplish MTS forecasting. Besides, we present theoretical analysis to demonstrate that the FGO is equivalent to graph convolutions in the time domain, which further explains the validity of FourierGNN.

Finally, we perform extensive experiments on seven real-world benchmarks. Experimental results demonstrate that FourierGNN achieves an average of more than 10% improvement in accuracy compared with state-of-the-art methods. In addition, FourierGNN achieves higher forecasting efficiency, which has about 14.6% less costs in training time and 20% less parameter volumes, compared with most lightweight GNN-based forecasting methods.

## 2 Related Work

**Graph Neural Networks for Multivariate Time Series Forecasting**   Multivariate time series (MTS) have embraced GNN due to their best capability of modeling structural dependencies between variates [17, 2, 13, 12, 11, 18, 19]. Most of these models, such as STGCN [12], DCRNN [18],

and TAMP-S2GCNets [11], require a pre-defined graph structure which is usually unknown in most cases. For this limitation, some studies enable to automatically learn the graphs by the inter-series correlations, e.g., by node similarity [20, 2, 17] or self-attention mechanism [10]. However, these methods always adopt a *graph network* for spatial correlations and a *temporal network* for temporal dependencies separately [20, 17, 10, 2]. For example, AGCRN [2] use a GCN [14] and a GRU [21], GraphWaveNet [17] use a GCN and a TCN [8], etc. In this paper, we propose an unified spatiotemporal formulation with pure graph networks for MTS forecasting.

**Multivariate Time Series Forecasting with Fourier Transform**    Recently, many MTS forecasting models have integrated the Fourier theory into deep neural networks [22, 23]. For instance, SFM [24] decomposes the hidden state of LSTM into multiple frequencies by Discrete Fourier Transform (DFT). mWDN [25] decomposes the time series into multilevel sub-series by discrete wavelet decomposition (DWT) and feeds them to LSTM network. ATFN [26] proposes a Discrete Fourier Transform-based block to capture dynamic and complicated periodic patterns of time series data. FEDformer [27] proposes Discrete Fourier Transform-based attention mechanism with low-rank approximation in frequency. While these models only capture temporal dependencies with Fourier Transform, StemGNN [10] takes the advantages of both spatial correlations and temporal dependencies in the spectral domain by utilizing Graph Fourier Transform (GFT) to perform graph convolutions and Discrete Fourier Transform (DFT) to calculate the series relationships.

## 3    Problem Definition

Given the multivariate time series input, i.e., the lookback window $X_t = [\boldsymbol{x}_{t-T+1}, ..., \boldsymbol{x}_t] \in \mathbb{R}^{N \times T}$ at timestamps $t$ with the number of series (variates) $N$ and the lookback window size $T$, where $\boldsymbol{x}_t \in \mathbb{R}^N$ denotes the multivariate values of $N$ series at timestamp $t$. Then, the *multivariate time series forecasting* task is to predict the next $\tau$ timestamps $Y_t = [\boldsymbol{x}_{t+1}, ..., \boldsymbol{x}_{t+\tau}] \in \mathbb{R}^{N \times \tau}$ based on the historical $T$ observations $X_t = [\boldsymbol{x}_{t-T+1}, ..., \boldsymbol{x}_t]$. The forecasting process can be given by:

$$\hat{Y}_t := F_\theta(X_t) = F_\theta([\boldsymbol{x}_{t-T+1}, ..., \boldsymbol{x}_t]) \tag{1}$$

where $\hat{Y}_t$ are the predictions corresponding to the ground truth $Y_t$. The forecasting function is denoted as $F_\theta$ parameterized by $\theta$. In practice, many MTS forecasting models usually leverage a *graph network* (assume parameterized by $\theta_g$) to learn the spatial dynamics and a *temporal network* (assume parameterized by $\theta_t$) to learn the temporal dependencies, respectively [17, 10, 2, 13, 11]. Thus, the original definition of Equation (1) can be rewritten to:

$$\hat{Y}_t := F_{\theta_g, \theta_t}(X_t) = F_{\theta_g, \theta_t}([\boldsymbol{x}_{t-T+1}, ..., \boldsymbol{x}_t]) \tag{2}$$

where original parameters $\theta$ are exposed to the parameters of the graph network $\theta_g$ and the temporal network $\theta_t$ to make prediction based on the learned spatial-temporal dependencies.

## 4    Methodology

In this section, we elaborate on our proposed framework: First, we start with our pure graph formulation with a novel hypervariate graph structure for MTS forecasting in Section 4.1. Then, we illustrate the proposed neural architecture, Fourier Graph Neural Network (FourierGNN), for this formulation in Section 4.2. Besides, we theoretically analyze FourierGNN to demonstrate its architecture validity, and also conduct complexity analysis to show its efficiency. Finally, we introduce certain inductive bias to instantiate FourierGNN for MTS forecasting in Section 4.3.

### 4.1    The Pure Graph Formulation

To overcome the uncertain compatibility of the graph network and the temporal network as afore-mentioned in Section 1, and learn the united spatiotemporal dynamics, we propose a *pure* graph formulation that refines Equation (2) by a novel data structure, *hypervariate graph*, for time series.

**Definition 1** (**Hypervariate Graph**).  *Given a multivariate time series window as input $X_t \in \mathbb{R}^{N \times T}$ of $N$ variates at timestamp $t$, we construct a hypervariate graph of $NT$ nodes, $\mathcal{G}_t = (X_t^{\mathcal{G}}, A_t^{\mathcal{G}})$, by regarding each element of $X_t$ as one node of $\mathcal{G}_t$ such that $X_t^{\mathcal{G}} \in \mathbb{R}^{NT \times 1}$ stands for the node feature and $A_t^{\mathcal{G}} \in \mathbb{R}^{NT \times NT}$ is the adjacency matrix initialized to make $\mathcal{G}_t$ as a fully-connected graph.*

Since the prior graph structure is usually unknown in most multivariate time series scenarios [10, 2, 13], and the elements of $X_t$ are spatially or temporally correlated with each other because of time lag effect [28], we assume all nodes in the hypervariate graph $\mathcal{G}_t$ are fully-connected. The hypervariate graph $\mathcal{G}_t$ contains $NT$ nodes representing the values of each variate at each timestamp in $X_t$, which can learn a high-resolution representation across timestamps and variates (more explanations of the hypervariate graph can be seen in Appendix C.1). We present an example hypervariate graph of three time series in Figure 1. Thus, with such a data structure, we can reformulate the *multivariate*

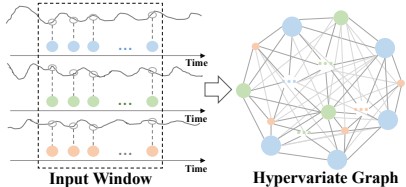

**Input Window**   **Hypervariate Graph**

Figure 1: Illustration of a hypervariate graph with three time series. Each value in the input window is considered as a node of the graph.

*time series forecasting* task into the predictions on the hypervariate graphs, and accordingly rewrite Equation (2) into:

$$\hat{Y}_t := F_{\theta_{\mathcal{G}}}(X_t^{\mathcal{G}}, A_t^{\mathcal{G}}) \tag{3}$$

where $\theta_{\mathcal{G}}$ stands for the network parameters for hypervariate graphs. With such a formulation, we can view the spatial dynamics and the temporal dependencies from an united perspective, which benefits modeling the real-world spatiotemporal inter-dependencies.

## 4.2 FourierGNN

Though the pure graph formulation can enhance spatiotemporal modeling, the order and size of hypervariate graphs increase with the number of variates $N$ and the size of window $T$, which makes classic graph networks (e.g., GCN [14] and GAT [15]) computationally expensive (usually quadratic complexity) and suffer from optimization difficulty in obtaining accurate hidden node representations [16]. In this regard, we propose an efficient and effective method, FourierGNN, for the pure graph formulation. The main architecture of FourierGNN is built upon our proposed Fourier Graph Operator (FGO), a learnable network layer in Fourier space, which is detailed as follows.

**Definition 2 (Fourier Graph Operator).** *Given a graph $G = (X, A)$ with node features $X \in \mathbb{R}^{n \times d}$ and the adjacency matrix $A \in \mathbb{R}^{n \times n}$, where $n$ is the number of nodes and $d$ is the number of features, we introduce a weight matrix $W \in \mathbb{R}^{d \times d}$ to acquire a tailored Green's kernel $\kappa : [n] \times [n] \to \mathbb{R}^{d \times d}$ with $\kappa[i, j] := A_{ij} \circ W$ and $\kappa[i, j] = \kappa[i - j]$. We define $\mathcal{S}_{A,W} := \mathcal{F}(\kappa) \in \mathbb{C}^{n \times d \times d}$ as a Fourier Graph Operator (FGO), where $\mathcal{F}$ denotes Discrete Fourier Transform (DFT).*

According to the convolution theorem [29] (see Appendix B), we can write the multiplication between $\mathcal{F}(X)$ and FGO $\mathcal{S}_{A,W}$ in Fourier space as:

$$\mathcal{F}(X)\mathcal{F}(\kappa) = \mathcal{F}((X * \kappa)[i]) = \mathcal{F}(\sum_{j=1}^{n} X[j]\kappa[i - j]) = \mathcal{F}(\sum_{j=1}^{n} X[j]\kappa[i, j]), \qquad \forall i \in [n] \tag{4}$$

where $(X * \kappa)[i]$ denotes the convolution of $X$ and $\kappa$. As defined $\kappa[i, j] = A_{ij} \circ W$, it yields $\sum_{j=1}^{n} X[j]\kappa[i, j] = \sum_{j=1}^{n} A_{ij}X[j]W = AXW$. Accordingly, we can get the convolution equation:

$$\mathcal{F}(X)\mathcal{S}_{A,W} = \mathcal{F}(AXW). \tag{5}$$

In particular, turning to our case of the fully-connected hypervariate graph, we can adopt a $n$-invariant FGO $\mathcal{S} \in \mathbb{C}^{d \times d}$ that has a computationally low cost compared to previous $\mathbb{C}^{n \times d \times d}$. We provide more details and explanations in Appendix C.2.

From Equation (5), we can observe that performing the multiplication between $\mathcal{F}(X)$ and FGO $\mathcal{S}$ in Fourier space corresponds to a graph shift operation (i.e., a graph convolution) in the time domain [20]. Since the multiplications in Fourier space ($\mathcal{O}(n)$) have much lower complexity than the above shift operations ($\mathcal{O}(n^2)$) in the time domain (See *Complexity Analysis* below), it motivates us to develop a highly efficient graph neural network in Fourier space.

To this end, we propose the *Fourier Graph Neural Networks* (FourierGNN) based on FGO. Specifically, by stacking multiple layers of FGOs, we can define the $K$-layer Fourier graph neural networks given a graph $G = (X, A)$ with node features $X \in \mathbb{R}^{n \times d}$ and the adjacency matrix $A \in \mathbb{R}^{n \times n}$ as:

$$\text{FourierGNN}(X, A) := \sum_{k=0}^{K} \sigma(\mathcal{F}(X)\mathcal{S}_{0:k} + b_k), \quad \mathcal{S}_{0:k} = \prod_{i=0}^{k} \mathcal{S}_i. \tag{6}$$

Herein, $\mathcal{S}_k$ is the FGO in the $k$-th layer, satisfying $\mathcal{F}(X)\mathcal{S}_k = \mathcal{F}(A_k X W_k)$ with $W_k \in \mathbb{R}^{d \times d}$ being the weights and $A_k \in \mathbb{R}^{n \times n}$ corresponding to the $k$-th adjacency matrix sharing the same sparsity pattern of $A$, and $b_k \in \mathbb{C}^d$ are the complex-valued biases parameters; $\mathcal{F}$ stands for Discrete Fourier Transform; $\sigma$ is the activation function. In particular, $\mathcal{S}_0, W_0, A_0$ are the identity matrix, and we adopt identical activation at $k = 0$ to obtain residual $\mathcal{F}(X)$. All operations in FourierGNN are performed in Fourier space. Thus, all parameters, i.e., $\{\mathcal{S}_k, b_k\}_{k=1}^{K}$, are complex numbers.

The core operation of FourierGNN is the summation of recursive multiplications with nonlinear activation functions. Specifically, the recursive multiplications between $\mathcal{F}(X)$ and $\mathcal{S}$, i.e., $\mathcal{F}(X)\mathcal{S}_{0:k}$, are equivalent to the multi-order convolutions on the graph structure (see *Theoretical Analysis* below). Nonlinear activation functions $\sigma$ are introduced to address the capability limitations of modeling nonlinear information diffusion on graphs in the summation.

**Theoretical Analysis**   We theoretically analyze the effectiveness and interpretability of FourierGNN and verify the validity of its architecture. For convenience, we exclude the non-linear activation function $\sigma$ and learnable bias parameters $b$ from Equation (6), and focus on $\mathcal{F}(X)\mathcal{S}_{0:k}$.

**Proposition 1.** *Given a graph $G = (X, A)$ with node features $X \in \mathbb{R}^{n \times d}$ and adjacency matrix $A \in \mathbb{R}^{n \times n}$, the recursive multiplication of FGOs in Fourier space is equivalent to multi-order convolutions in the time domain:*

$$\mathcal{F}^{-1}(\mathcal{F}(X)\mathcal{S}_{0:k}) = A_{k:0} X W_{0:k}, \quad \mathcal{S}_{0:k} = \prod_{i=0}^{k} \mathcal{S}_i, A_{k:0} = \prod_{i=k}^{0} A_i, W_{0:k} = \prod_{i=0}^{k} W_i \quad (7)$$

*where $A_0, \mathcal{S}_0, W_0$ are the identity matrix, $A_k \in \mathbb{R}^{n \times n}$ corresponds to the $k$-th diffusion step sharing the same sparsity pattern of $A$, $W_k \in \mathbb{R}^{d \times d}$ is the $k$-th weight matrix, $\mathcal{S}_k \in \mathbb{C}^{d \times d}$ is the $k$-th FGO satisfying $\mathcal{F}(A_k X W_k) = \mathcal{F}(X)\mathcal{S}_k$, and $\mathcal{F}$ and $\mathcal{F}^{-1}$ denote DFT and its inverse, respectively.*

In the time domain, operation $A_{k:0} X W_{0:k}$ adopts different weights $W_k \in \mathbb{R}^{d \times d}$ to weigh the information of different neighbors in different diffusion orders, beneficial to capture the extensive dependencies on graphs [20, 30, 31]. This indicates FourierGNN is expressive in modeling the complex correlations among graph nodes, i.e., spatiotemporal dependencies in the hypervariate graph. The proof of Proposition 1 and more explanations of FourierGNN are provided in Appendix C.3.

**Complexity Analysis**   The time complexity of $\mathcal{F}(X)\mathcal{S}$ is $\mathcal{O}(nd \log n + nd^2)$, which includes the Discrete Fourier Transform (DFT), the Inverse Discrete Fourier Transform (IDFT), and the matrix multiplication in the Fourier space. Comparatively, the time complexity of the equivalent operations of $\mathcal{F}(X)\mathcal{S}$ in the time domain, i.e., $AXW$, is $\mathcal{O}(n^2 d + nd^2)$. Then, as a $K$-order summation of a recursive multiplication of $\mathcal{F}(X)\mathcal{S}$, FourierGNN, achieves the time complexity of $\mathcal{O}(nd \log n + Knd^2)$, including DFT and IDFT, and the recursive multiplication of FGOs. Overall, the Log-linear $\mathcal{O}(n \log n)$ complexity makes FourierGNN much more efficient.

**FourierGNN vs Other Graph Networks**   We analyze the connection and difference between our FourierGNN with GCN [14] and GAT [15]. From the complexity perspective, FourierGNN with log-linear complexity shows much higher efficiency than GCN and GAT. Regarding the network architecture, we analyze them from two main perspectives: (1) Domain. GAT implements operations in the time domain, while GCN and FourierGNN are in Fourier space. However, GCN achieves the transformation through the Graph Fourier Transform (GFT), whereas FourierGNN utilizes the Discrete Fourier Transform (DFT). (2) Information diffusion: GAT aggregates neighbor nodes with varying weights to via attention mechanisms. FourierGNN and GCN update node information via convoluting neighbor nodes. Different from GCN, FourierGNN assigns varying importance to neighbor nodes in different diffusion steps. We provide a detailed comparison in Appendix D.

### 4.3   Multivariate Time Series Forecasting with FourierGNN

In this section, we instantiate FourierGNN for MTS forecasting. The overall architecture of our model is illustrated in Figure 2. Given the MTS input data $X_t \in \mathbb{R}^{N \times T}$, first we construct a fully-connected *hypervariate graph* $\mathcal{G}_t = (X_t^{\mathcal{G}}, A_t^{\mathcal{G}})$ with $X_t^{\mathcal{G}} \in \mathbb{R}^{NT \times 1}$ and $A_t^{\mathcal{G}} \in \{1\}^{n \times n}$. Then, we project $X_t^{\mathcal{G}}$ into node embeddings $\mathbf{X}_t^{\mathcal{G}} \in \mathbb{R}^{NT \times d}$ by assigning a $d$-dimension vector for each node using an embedding matrix $E_\phi \in \mathbb{R}^{1 \times d}$, i.e., $\mathbf{X}_t^{\mathcal{G}} = X_t^{\mathcal{G}} \times E_\phi$.

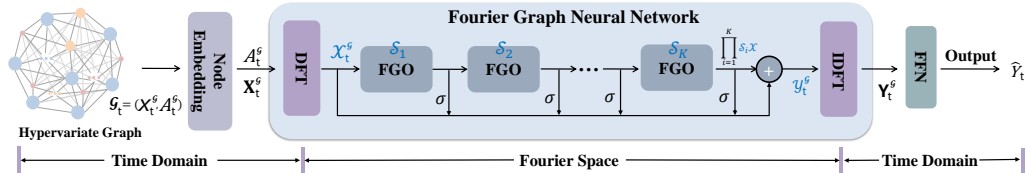

Figure 2: The network architecture of MTS forecasting with FourierGNN (blue characters denote complex values, such as $\mathcal{X}_t^{\mathcal{G}}$ $\mathcal{S}_i$). Given the hypervariate graph $\mathcal{G} = (X_t^{\mathcal{G}}, A_t^{\mathcal{G}})$, we 1) embed nodes of $X_t^{\mathcal{G}} \in \mathbb{R}^{NT \times 1}$ to obtain node embeddings $\mathbf{X}_t^{\mathcal{G}} \in \mathbb{R}^{NT \times d}$; 2) feed embedded hypervariate graphs to FourierGNN: (i) transform $\mathbf{X}_t^{\mathcal{G}}$ with DFT to $\mathcal{X}_t^{\mathcal{G}} \in \mathbb{C}^{NT \times d}$; (ii) conduct recursive multiplications and make summation to output $\mathcal{Y}_t^{\mathcal{G}}$; (iii) transform $\mathcal{Y}_t^{\mathcal{G}}$ back to time domain by IDFT, resulting in $\mathbf{Y}_t^{\mathcal{G}} \in \mathbb{R}^{NT \times d}$; 3) generate $\tau$-step predictions $\hat{Y}_t \in \mathbb{R}^{N \times \tau}$ via feeding $\mathbf{Y}_t^{\mathcal{G}}$ to fully-connected layers.

Subsequently, to capture the spatiotemporal dependencies simultaneously, we aim to feed multiple embeded hypervariate graphs with $\mathbf{X}_t^{\mathcal{G}}$ to FourierGNN. First, we perform Discrete Fourier Transform (DFT) $\mathcal{F}$ on each discrete spatio-temporal dimension of the embeddings $\mathbf{X}_t^{\mathcal{G}}$ and obtain the frequency output $\mathcal{X}_t^{\mathcal{G}} := \mathcal{F}(\mathbf{X}_t^{\mathcal{G}}) \in \mathbb{C}^{NT \times d}$. Then, we perform a recursive multiplication between $\mathcal{X}_t^{\mathcal{G}}$ and FGOs $\mathcal{S}_{0:k}$ in Fourier space and output the resulting representations $\mathcal{Y}_t^{\mathcal{G}}$ as:

$$\mathcal{Y}_t^{\mathcal{G}} = \text{FourierGNN}(\mathbf{X}_t^{\mathcal{G}}, A_t^{\mathcal{G}}) = \sum_{k=0}^{K} \sigma(\mathcal{F}(\mathbf{X}_t^{\mathcal{G}})\mathcal{S}_{0:k} + b_k), \quad \mathcal{S}_{0:k} = \prod_{i=0}^{k} \mathcal{S}_i. \tag{8}$$

Then $\mathcal{Y}_t^{\mathcal{G}}$ are transformed back to the time domain using Inverse Discrete Fourier Transform (IDFT) $\mathcal{F}^{-1}$, which yields $\mathbf{Y}_t^{\mathcal{G}} := \mathcal{F}^{-1}(\mathcal{Y}_t^{\mathcal{G}}) \in \mathbb{R}^{NT \times d}$.

Finally, according to the FourierGNN output $\mathbf{Y}_t^{\mathcal{G}}$ which encodes spatiotemporal inter-dependencies, we use two layer feed-forward networks (FFN) (see Appendix E.4 for more details) to project it onto $\tau$ future steps, resulting in $\hat{Y}_t = \text{FFN}(\mathbf{Y}_t^{\mathcal{G}}) \in \mathbb{R}^{N \times \tau}$.

## 5 Experiments

To evaluate the performance of FourierGNN, we conduct extensive experiments on seven real-world time series benchmarks to compare with state-of-the-art graph neural network-based methods.

### 5.1 Experimental Setup

**Datasets**. We evaluate our proposed method on seven representative datasets from various application scenarios, including traffic, energy, web traffic, electrocardiogram, and COVID-19. All datasets are normalized using the min-max normalization. Except the COVID-19 dataset, we split the other datasets into training, validation, and test sets with the ratio of 7:2:1 in a chronological order. For the COVID-19 dataset, the ratio is 6:2:2. More detailed information about datasets are in Appendix E.1.

**Baselines**. We conduct a comprehensive comparison of the forecasting performance between our FourierGNN and several representative and state-of-the-art (SOTA) models on the seven datasets, including classic method VAR [32], deep learning-based models such as SFM [24], LSTNet [33], TCN [8], DeepGLO [34], and CoST [36]. We also compare FourierGNN against GNN-based models like GraphWaveNet [17], StemGNN [10], MTGNN [13], and AGCRN [2], and two representative Transformer-based models like Reformer [35] and Informer [9], as well as two frequency enhanced Transformer-based models including Autoformer [4] and FEDformer [27]. In addition, we compare FourierGNN with SOTA models such as TAMP-S2GCNets [11], DCRNN [18], and STGCN [1], which require pre-defined graph structures. Please refer to Appendix E.2 for more implementation details of the adopted baselines.

**Experimental Settings**. All experiments are conducted in Python using Pytorch 1.8 [37] (except for SFM [24] which uses Keras) and performed on single NVIDIA RTX 3080 10G GPU. Our model is trained using RMSProp with a learning rate of $10^{-5}$ and MSE (Mean Squared Error) as the loss function. The best parameters for all comparative models are chosen through careful parameter tuning on the validation set. We use Mean Absolute Errors (MAE), Root Mean Squared Errors (RMSE), and

Table 1: Overall performance of forecasting models on the six datasets.

| Datasets | Solar | | | Wiki | | | Traffic | | |
|---|---|---|---|---|---|---|---|---|---|
| Models | MAE | RMSE | MAPE(%) | MAE | RMSE | MAPE(%) | MAE | RMSE | MAPE(%) |
| VAR [32] | 0.184 | 0.234 | 577.10 | 0.057 | 0.094 | 96.58 | 0.535 | 1.133 | 550.12 |
| SFM [24] | 0.161 | 0.283 | 362.89 | 0.081 | 0.156 | 104.47 | 0.029 | 0.044 | 59.33 |
| LSTNet [33] | 0.148 | 0.200 | 132.95 | 0.054 | 0.090 | 118.24 | 0.026 | 0.057 | **25.77** |
| TCN [8] | 0.176 | 0.222 | 142.23 | 0.094 | 0.142 | 99.66 | 0.052 | 0.067 | - |
| DeepGLO [34] | 0.178 | 0.400 | 346.78 | 0.110 | 0.113 | 119.60 | 0.025 | 0.037 | 33.32 |
| Reformer [35] | 0.234 | 0.292 | 128.58 | 0.048 | 0.085 | 73.61 | 0.029 | 0.042 | 112.58 |
| Informer [9] | 0.151 | 0.199 | 128.45 | 0.051 | 0.086 | 80.50 | 0.020 | 0.033 | 59.34 |
| Autoformer [4] | 0.150 | 0.193 | 103.79 | 0.069 | 0.103 | 121.90 | 0.029 | 0.043 | 100.02 |
| FEDformer [27] | 0.139 | 0.182 | **100.92** | 0.068 | 0.098 | 123.10 | 0.025 | 0.038 | 85.12 |
| GraphWaveNet [17] | 0.183 | 0.238 | 603 | 0.061 | 0.105 | 136.12 | 0.013 | 0.034 | 33.78 |
| StemGNN [10] | 0.176 | 0.222 | 128.39 | 0.190 | 0.255 | 117.92 | 0.080 | 0.135 | 64.51 |
| MTGNN [13] | 0.151 | 0.207 | 507.91 | 0.101 | 0.140 | 122.96 | 0.013 | 0.030 | 29.53 |
| AGCRN [2] | 0.123 | 0.214 | 353.03 | 0.044 | 0.079 | 78.52 | 0.084 | 0.166 | 31.73 |
| **FourierGNN** | **0.120** | **0.162** | 116.48 | **0.041** | **0.076** | **64.50** | **0.011** | **0.023** | 28.71 |

| Datasets | ECG | | | Electricity | | | COVID-19 | | |
|---|---|---|---|---|---|---|---|---|---|
| Models | MAE | RMSE | MAPE(%) | MAE | RMSE | MAPE(%) | MAE | RMSE | MAPE(%) |
| VAR [32] | 0.120 | 0.170 | 22.56 | 0.101 | 0.163 | 43.11 | 0.226 | 0.326 | 191.95 |
| SFM [24] | 0.095 | 0.135 | 24.20 | 0.086 | 0.129 | 33.71 | 0.205 | 0.308 | 76.08 |
| LSTNet [33] | 0.079 | 0.115 | 18.68 | 0.075 | 0.138 | 29.95 | 0.248 | 0.305 | 89.04 |
| TCN [8] | 0.078 | 0.107 | 17.59 | 0.057 | 0.083 | 26.64 | 0.317 | 0.354 | 151.78 |
| DeepGLO [34] | 0.110 | 0.163 | 43.90 | 0.090 | 0.131 | 29.40 | 0.169 | 0.253 | 75.19 |
| Reformer [35] | 0.062 | 0.090 | 13.58 | 0.078 | 0.129 | 33.37 | 0.152 | 0.209 | 132.78 |
| Informer [9] | 0.056 | 0.085 | 11.99 | 0.070 | 0.119 | 32.66 | 0.200 | 0.259 | 155.55 |
| Autoformer [4] | 0.055 | 0.081 | 11.37 | 0.056 | 0.083 | 25.94 | 0.159 | 0.211 | 136.24 |
| FEDformer [27] | 0.055 | 0.080 | 11.16 | 0.055 | 0.081 | 25.84 | 0.160 | 0.219 | 134.45 |
| GraphWaveNet [17] | 0.093 | 0.142 | 40.19 | 0.094 | 0.140 | 37.01 | 0.201 | 0.255 | 100.83 |
| StemGNN [10] | 0.100 | 0.130 | 29.62 | 0.070 | 0.101 | - | 0.421 | 0.508 | 141.01 |
| MTGNN [13] | 0.090 | 0.139 | 35.04 | 0.077 | 0.113 | 29.77 | 0.394 | 0.488 | 88.13 |
| AGCRN [2] | 0.055 | 0.080 | 11.75 | 0.074 | 0.116 | 26.08 | 0.254 | 0.309 | 83.37 |
| **FourierGNN** | **0.052** | **0.078** | **10.97** | **0.051** | **0.077** | **24.28** | **0.123** | **0.168** | **71.52** |

Table 2: Performance comparison under different prediction lengths on the COVID-19 dataset.

| Length | 3 | | | 6 | | | 9 | | | 12 | | |
|---|---|---|---|---|---|---|---|---|---|---|---|---|
| Metrics | MAE | RMSE | MAPE(%) | MAE | RMSE | MAPE(%) | MAE | RMSE | MAPE(%) | MAE | RMSE | MAPE(%) |
| GraphWaveNet [17] | 0.092 | 0.129 | 53.00 | 0.133 | 0.179 | 65.11 | 0.171 | 0.225 | 80.91 | 0.201 | 0.255 | 100.83 |
| StemGNN [10] | 0.247 | 0.318 | 99.98 | 0.344 | 0.429 | 125.81 | 0.359 | 0.442 | 131.14 | 0.421 | 0.508 | 141.01 |
| AGCRN [2] | 0.130 | 0.172 | 76.73 | 0.171 | 0.218 | 79.07 | 0.224 | 0.277 | 82.90 | 0.254 | 0.309 | 83.37 |
| MTGNN [13] | 0.276 | 0.379 | 91.42 | 0.446 | 0.513 | 133.49 | 0.484 | 0.548 | 139.52 | 0.394 | 0.488 | 88.13 |
| TAMP-S2GCNets [11] | 0.140 | 0.190 | **50.01** | 0.150 | 0.200 | **55.72** | 0.170 | 0.230 | 71.78 | 0.180 | 0.230 | **65.76** |
| CoST [36] | 0.122 | 0.246 | 68.74 | 0.157 | 0.318 | 72.84 | 0.183 | 0.364 | 77.04 | 0.202 | 0.377 | 80.81 |
| **FourierGNN(ours)** | **0.071** | **0.103** | 61.02 | **0.093** | **0.131** | 65.72 | **0.109** | **0.148** | 69.59 | **0.123** | **0.168** | 71.52 |

Mean Absolute Percentage Error (MAPE) to measure the performance. The evaluation details are in Appendix E.3 and more experimental settings are in Appendix E.4.

## 5.2 Main Results

We present the evaluation results with an input length of 12 and a prediction length of 12 in Table 1. Overall, FourierGNN achieves a new state-of-the-art on all datasets. On average, FourierGNN makes an improvement of 9.4% in MAE and 10.9% in RMSE compared to the best-performing across all datasets. Among these baselines, Reformer, Informer, Autoformer, and FEDformer are Transformer-based models that demonstrate competitive performance on Electricity and COVID-19 datasets, as they excel at capturing temporal dependencies. However, they have limitations in capturing the spatial dependencies explicitly. GraphWaveNet, MTGNN, StemGNN, and AGCRN are GNN-based models that show promising results on Wiki, Traffic, Solar, and ECG datasets, primarily due to their capability to handle spatial dependencies among variates. However, they are limited in their capacity to simultaneously capture spatiotemporal dependencies. FourierGNN outperforms the baseline models since it can learn comprehensive spatiotemporal dependencies simultaneously and attends to time-varying dependencies among variates.

**Multi-Step Forecasting** To further evaluate the performance in multi-step forecasting, we compare FourierGNN with other GNN-based MTS models (including StemGNN [10], AGCRN [2],

GraphWaveNet [17], MTGNN [13], and TAMP-S2GCNets [11]) and a representation learning model (CoST [36]) on COVID-19 dataset under different prediction lengths, and the results are shown in Table 2. It shows that FourierGNN achieves an average 30.1% and 30.2% improvement on MAE and RMSE respectively over the best baseline. In Appendix F, we include more experiments and analysis under different prediction lengths, and further compare FourierGNN with models that require pre-defined graph structures.

## 5.3 Model Analysis

**Efficiency Analysis** We investigate the parameter volumes and training time costs of FourierGNN, StemGNN [10], AGCRN [2], GraphWaveNet [17], and MTGNN [13] on two representative datasets, including the Wiki dataset and the Traffic dataset. The results are reported in Table 3, showing the comparison of parameter volumes and average time costs over five rounds of experiments. In terms of parameters, FourierGNN exhibits the lowest volume of parameters among the comparative models. Specifically, it achieves a reduction of 32.2% and 9.5% in parameters compared to GraphWaveNet on Traffic and Wiki datasets, respectively. This reduction is mainly attributed that FourierGNN has shared scale-free parameters for each node. Regarding training time, FourierGNN runs much faster than all baseline models, and it demonstrates efficiency improvements of 5.8% and 23.3% over the fast baseline GraphWaveNet on Traffic and Wiki datasets, respectively. Considering variate number of Wiki dataset is about twice larger than that of Traffic dataset, FourierGNN exhibits larger efficiency superiority with the baselines. These findings highlight the high efficiency of FourierGNN in computing graph operations and its scalability to large datasets with extensive graphs, which is important for the pure graph formulation due to the larger size of hypervariate graphs with $NT$ nodes.

Table 3: Comparisons of parameter volumes and training time costs on datasets Traffic and Wiki.

| Models | Traffic | | Wiki | |
|---|---|---|---|---|
| | Parameters | Training (s/epoch) | Parameters | Training (s/epoch) |
| StemGNN | $1,606,140$ | $185.86\pm2.22$ | $4,102,406$ | $92.95\pm1.39$ |
| MTGNN | $707,516$ | $169.34\pm1.56$ | $1,533,436$ | $28.69\pm0.83$ |
| AGCRN | $749,940$ | $113.46\pm1.91$ | $755,740$ | $22.48\pm1.01$ |
| GraphWaveNet | $280,860$ | $105.38\pm1.24$ | $292,460$ | $21.23\pm0.76$ |
| FourierGNN | $190,564$ | $99.25\pm1.07$ | $264,804$ | $16.28\pm0.48$ |

**Ablation Study** We perform an ablation study on the METR-LA dataset to assess the individual contributions of different components in FourierGNN. The results, presented in Table 4, validate the effectiveness of each component. Specifically, **w/o Embedding** emphasizes the significance of performing node embedding to improve model generalization. **w/o Dynamic FGO** using the same FGO verifies the effectiveness of applying different FGOs in capturing time-varying dependencies. In addition, **w/o Residual** represents FourierGNN without the $K = 0$ layer, while **w/o Summation** adopts the last order (layer) output, i.e., $\mathcal{X}S_{0:K}$, as the output of FourierGNN. These results demonstrate the importance of high-order diffusion and the contribution of multi-order diffusion. More results and analysis of the ablation study are provided in Appendix G.3.

Table 4: Ablation study on METR-LA dataset.

| metrics | w/o Embedding | w/o Dynamic FGO | w/o Residual | w/o Summation | FourierGNN |
|---|---|---|---|---|---|
| MAE | 0.053 | 0.055 | 0.054 | 0.054 | 0.050 |
| RMSE | 0.116 | 0.114 | 0.115 | 0.114 | 0.113 |
| MAPE(%) | 86.73 | 86.69 | 86.75 | 86.62 | 86.30 |

## 5.4 Visualization

To gain a better understanding of the hypervariate graph and FourierGNN in spatiotemporal modeling for MTS forecasting, We conduct visualization experiments on the METR-LA and COVID-19 datasets. Please refer to Appendix E.5 for more information on the visualization techniques used.

**Visualization of temporal representations learned by FourierGNN** In order to showcase the temporal dependencies learning capability of FourierGNN, we visualize the temporal adjacency matrix of different variates. Specifically, we randomly select 8 counties from the COVID-19 dataset and calculate the relations of 12 consecutive time steps for each county. Then, we visualize the

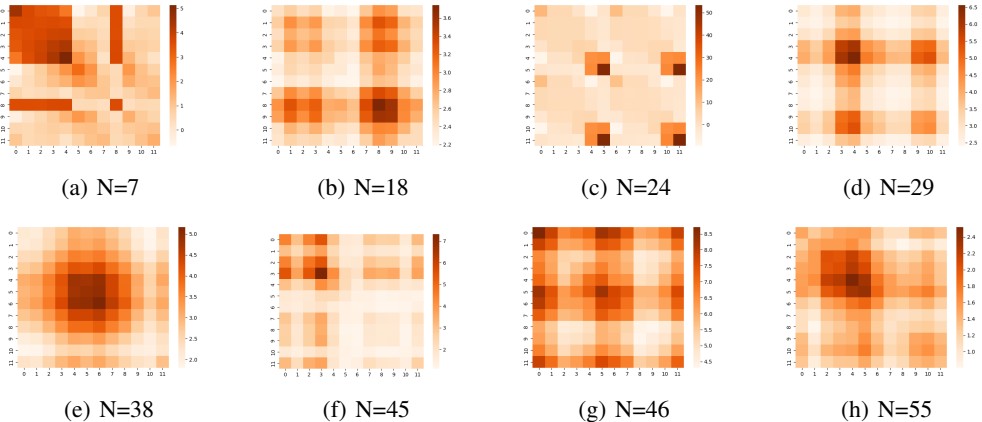

| (a) N=7 | (b) N=18 | (c) N=24 | (d) N=29 |
|---|---|---|---|
| (e) N=38 | (f) N=45 | (g) N=46 | (h) N=55 |

Figure 3: The temporal adjacency matrix of eight variates on COVID-19 dataset.

adjacency matrix by a heatmap, and the results are illustrated in Figure 3, where $N$ denotes the index of the country (variate). It shows that FourierGNN learns distinct temporal patterns for each county, indicating that the hypervariate graph can encode rich and discriminative temporal dependencies.

**Visualization of spatial representations learned by FourierGNN**    To investigate the spatial correlations learning capability of FourierGNN, we visualize the generated adjacency matrix based on the representations learned by FourierGNN on the METR-LA dataset. Specifically, we randomly select 20 detectors and visualize their corresponding adjacency matrix via a heatmap, as depicted in Figure 4.

By examining the adjacency matrix in conjunction with the actual road map, we observe: 1) the detectors (7, 8, 9, 11, 13, 18) are very close w.r.t. the physical distance, corresponding to the high values of their correlations with each other in the heatmap; 2) the detectors 4, 14 and 16 have small overall correlation values since they are far from other detectors; 3) however, compared with detectors 14 and 16, the detector 4 has slightly higher correlation values to other detectors, e.g., 7, 8, 9, which is because although they are far apart, the detectors 4, 7, 8, 9 are on the same road. The results verify that the hypervariate graph structure can represent highly interpretative correlations.

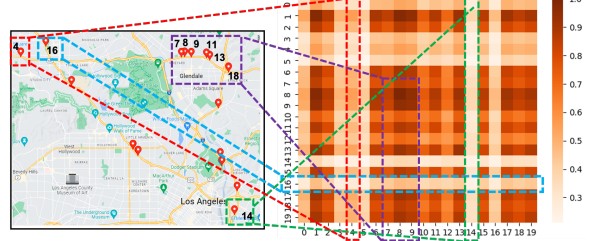

Figure 4: The adjacency matrix (right) learned by FourierGNN and the corresponding road map (left).

Moreover, to gain a understanding of how FGO works, we visualize the output of each layer of FourierGNN, and the visualization results demonstrate that FGO can adaptively and effectively capture important patterns while removing noises to a learn discriminative model. Further details can be found in Appendix H.1. Additionally, to investigate the ability of FourierGNN to capture time-varying dependencies among variates, we further visualize the spatial correlations at different timestamps. The results illustrate that FourierGNN can effectively attend to the temporal variability in the data. For more information, please refer to Appendix H.2.

## 6   Conclusion Remarks

In this paper, we explore an interesting direction of directly applying graph networks for MTS forecasting from a pure graph perspective. To overcome the previous separate spatial and temporal modeling problem, we build a hypervariate graph, regarding each series value as a graph node, which considers spatiotemporal dynamics unitedly. Then, we formulate time series forecasting on the hypervariate graph and propose FourierGNN by stacking Fourier Graph Operator (FGO) to perform matrix multiplications in Fourier space, which can accommodate adequate learning expressiveness with much lower complexity. Extensive experiments demonstrate that FourierGNN achieves state-of-the-art performances with higher efficiency and fewer parameters, and the hypervariate graph structure exhibits strong capabilities to encode spatiotemporal inter-dependencies.

## Acknowledgments and Disclosure of Funding

The work was supported in part by the National Key Research and Development Program of China under Grant 2020AAA0104903 and 2019YFB1406300, and National Natural Science Foundation of China under Grant 62072039 and 62272048.

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

# A Notation

Table 5: Notations

| | |
|---|---|
| $X_t$ | multivariate time series input at timestamps $t$, $X \in \mathbb{R}^{N \times T}$ |
| $x_t$ | the multivariate values of $N$ series at timestamp t, $x_t \in \mathbb{R}^N$ |
| $Y_t$ | the next $\tau$ timestamps of multivariate time series, $Y_t \in \mathbb{R}^{N \times \tau}$ |
| $\hat{Y}_t$ | the prediction values of multivariate time series for next $\tau$ timestamps, $\hat{Y}_t \in \mathbb{R}^{N \times \tau}$ |
| $N$ | the number of series |
| $T$ | the lookback window size |
| $\tau$ | the prediction length of multivariate time series forecasting |
| $\mathcal{G}_t$ | the hypervariate graph, $\mathcal{G}_t = \{X_t^{\mathcal{G}}, A_t^{\mathcal{G}}\}$ attributed to $X_t^{\mathcal{G}}$ |
| $X_t^{\mathcal{G}}$ | the nodes of the hypervariate graph, $X_t^{\mathcal{G}} \in \mathbb{R}^{NT \times 1}$ |
| $A_t^{\mathcal{G}}$ | the adjacency matrix of $\mathcal{G}_t$, $A_t^{\mathcal{G}} \in \mathbb{R}^{NT \times NT}$ |
| $\mathcal{S}$ | the Fourier Graph Operator |
| $d$ | the embedding dimension |
| $\mathbf{X}_t^{\mathcal{G}}$ | the embedding of $X_t^{\mathcal{G}}$, $\mathbf{X}_t^{\mathcal{G}} \in \mathbb{R}^{NT \times d}$ |
| $\mathcal{X}_t^{\mathcal{G}}$ | the spectrum of $\mathbf{X}_t^{\mathcal{G}}$, $\mathcal{X}_t^{\mathcal{G}} \in \mathbb{C}^{NT \times d}$ |
| $\mathcal{Y}_t^{\mathcal{G}}$ | the output of FourierGNN, $\mathcal{Y}_t^{\mathcal{G}} \in \mathbb{C}^{NT \times d}$ |
| $\theta_g$ | the parameters of the graph network |
| $\theta_t$ | the parameters of the temporal network |
| $\theta_{\mathcal{G}}$ | the network parameters for hypervariate graphs |
| $E_\phi$ | the embedding matrix, $E_\phi \in \mathbb{R}^{1 \times d}$ |
| $\kappa$ | the kernel function |
| $W$ | the weight matrix |
| $b$ | the complex bias weights |
| $\mathcal{F}$ | Discrete Fourier Transform |
| $\mathcal{F}^{-1}$ | Inverse Discrete Fourier Transform |
| F | the forecasting model |

# B Convolution Theorem

The convolution theorem [29] is one of the most important properties of the Fourier transform. It states the Fourier transform of a convolution of two signals equals the pointwise product of their Fourier transforms in the frequency domain. Given a signal $x[n]$ and a filter $h[n]$, the convolution theorem can be defined as follows:

$$\mathcal{F}((x * h)[n]) = \mathcal{F}(x)\mathcal{F}(h) \tag{9}$$

where $(x * h)[n] = \sum_{m=0}^{N-1} h[m]x[(n-m)_N]$ denotes the convolution of $x$ and $h$, $(n-m)_N$ denotes $(n-m)$ modulo N, and $\mathcal{F}(x)$ and $\mathcal{F}(h)$ denote discrete Fourier transform of $x[n]$ and $h[n]$, respectively.

## C Explanations and Proofs

### C.1 The Explanations of the Hypervariate Graph Structure

Note that the time lag effect between time-series variables is a common phenomenon in real-world multivariate time series scenarios, for example, the time lag influence between two financial assets (e.g. dollar and gold) of a portfolio. It is beneficial but challenging to consider dependencies between different variables under different timestamps.

The hypervariate graph connecting any two variables at any two timestamps aims to encode high-resolution spatiotemporal dependencies. It embodies not only the intra-series temporal dependencies (node connections of each individual variable), inter-series spatial dependencies (node connections under each single time step), and also the time-varying spatiotemporal dependencies (node connections between different variables at different time steps). By leveraging the hypervariate graph structure, we can effectively learn the spatial and temporal dependencies. This approach is distinct from previous methods that represent the spatial and temporal dependencies separately using two network structures.

### C.2 The Interpretation of $n$-invariant FGO

**Why $\mathcal{F}(\kappa) \in \mathbb{C}^{n \times d \times d}$?** From Definition 2, we know that the kernel $\kappa$ is defined as a matrix-valued projection, i.e., $\kappa : [n] \times [n] \to \mathbb{R}^{d \times d}$. Note that we assume $\kappa$ is in the special case of the Green's kernel, i.e., a translation-invariant kernel $\kappa[i, j] = \kappa[i - j]$. Accordingly, $\kappa$ can be reduced: $\kappa : [n] \to \mathbb{R}^{d \times d}$ where we can parameterize $\mathcal{F}(\kappa)$ with a complex-valued matrix $\mathbb{C}^{n \times d \times d}$.

**What is $n$-invariant FGO?** Turning to our case of the fully-connected hypervariate graph, we can consider a special case of $\kappa$, i.e., a space-invariant kernel $\kappa[i, j] = \kappa[\varrho]$ with $\varrho$ being a constant scalar. Accordingly, we can parameterize FGO $\mathcal{S}$ with a $n$-invariant complex-valued matrix $\mathbb{C}^{d \times d}$.

**The interpretation of $n$-invariant FGO.** An $n$-invariant FGO is similar to a shared-weight convolution kernel or filter of CNNs that slide along ($[n] \times [n]$) input features, which effectively reduces parameter volumes and saves computation costs. Note that although we adopt the same transformation (i.e., the $n$-invariant FGO) over $NT$ frequency points, we embed the raw MTS inputs in the $d$-dimension distributive space beforehand and then perform FourierGNN over MTS embeddings, which can be analogized as $d$ convolution kernels/filters in each convolutional layer in CNNs. This can ensure FourierGNN is able to learn informative features/patterns to improve its model capacity (See the following analysis of the effectiveness of $n$-invariant FGO).

**The effectiveness of $n$-invariant FGO.** In addition, the $n$-invariant parameterized FGO is empirically proven effective to improve model generalization and achieve superior forecasting performance (See the ablation study in Section 5.3 for more details). Although parameterizing $\mathcal{F}(\kappa) \in \mathbb{C}^{n \times d \times d}$ (i.e., an $n$-variant FGO) may be more powerful and flexible than the $n$-invariant FGO in terms of forecasting performance, it introduces much more parameters and training time costs, especially in case of multi-layer FourierGNN, and may obtain inferior performance due to inadequate training or overfitting. As indicated in Table 6, the FourierGNN with the $n$-invariant FGO achieves slightly better performance than that with the $n$-variant FGO on ECG and COVID-19, respectively. Notably, the FourierGNN with the $n$-variant FGO introduces a much larger parameter volume proportional to $n$ and requires significantly more training time. In contrast, $n$-invariant FGO is $n$-agnostic and lightweight, which is a more wise and efficient alternative. These results confirm our design and verify the effectiveness and applicability of $n$-invariant FGO.

Table 6: Comparison between FourierGNN models with $n$-invariant FGO and $n$-variant FGO on the ECG and COVID-19 datasets.

| Datasets | Models | Parameters (M) | Training (s/epoch) | MAE | RMSE | MAPE (%) |
|---|---|---|---|---|---|---|
| ECG | $n$-invariant | **0.18** | **12.45** | **0.052** | 0.078 | **10.97** |
| | $n$-variant | 82.96 | 104.06 | 0.053 | 0.078 | 11.05 |
| COVID-19 | $n$-invariant | **1.06** | **0.62** | **0.123** | **0.168** | **71.52** |
| | $n$-variant | 130.99 | 7.46 | 0.129 | 0.174 | 72.12 |

## C.3 Proof of Proposition 1 and Interpretation of FourierGNN

**Proposition 1.** *Given a graph $G = (X, A)$ with node features $X \in \mathbb{R}^{n \times d}$ and adjacency matrix $A \in \mathbb{R}^{n \times n}$, the recursive multiplication of FGOs in Fourier space is equivalent to multi-order convolutions in the time domain:*

$$\mathcal{F}^{-1}(\mathcal{F}(X)\mathcal{S}_{0:k}) = A_{k:0}XW_{0:k}, \quad \mathcal{S}_{0:k} = \prod_{i=0}^{k} \mathcal{S}_i, A_{k:0} = \prod_{i=k}^{0} A_i, W_{0:k} = \prod_{i=0}^{k} W_i$$

*where $A_0, \mathcal{S}_0, W_0$ are the identity matrix, $A_k \in \mathbb{R}^{n \times n}$ corresponds to the $k$-th diffusion step sharing the same sparsity pattern of $A$, $W_k \in \mathbb{R}^{d \times d}$ is the $k$-th weight matrix, $\mathcal{S}_k \in \mathbb{C}^{d \times d}$ is the $k$-th FGO satisfying $\mathcal{F}(A_kXW_k) = \mathcal{F}(X)\mathcal{S}_k$, and $\mathcal{F}$ and $\mathcal{F}^{-1}$ denote DFT and its inverse, respectively.*

*Proof.* The proof aims to demonstrate the equivalence between the recursive multiplication of FGOs in Fourier space and multi-order convolutions in the time domain. According to $\mathcal{F}(A_kXW_k) = \mathcal{F}(X)\mathcal{S}_k$, we expand the multi-order convolutions $A_{0:K}XW_{0:K}$ in the time domain using a set of FGOs in Fourier space:

$$
\begin{aligned}
\mathcal{F}(A_K A_{K-1} \cdots A_0 X W_0 \cdots W_{K-1} W_K) &= \mathcal{F}(A_K(A_{K-1}...A_0XW_0 \cdots W_{K-1})W_K) \\
&= \mathcal{F}(A_{K-1}...A_0XW_0 \cdots W_{K-1})\mathcal{S}_K \\
&= \mathcal{F}(A_{K-1}(A_{K-2}...A_0XW_0 \cdots W_{K-2})W_{K-1})\mathcal{S}_K \\
&= \mathcal{F}(A_{K-2}...A_0XW_0 \cdots W_{K-2})\mathcal{S}_{K-1}\mathcal{S}_K \\
&= \cdots \\
&= \mathcal{F}(X)\mathcal{S}_0 \cdots \mathcal{S}_{K-1}\mathcal{S}_K \\
&= \mathcal{F}(X)\mathcal{S}_{0:K}
\end{aligned}
\tag{10}
$$

where it yields $\mathcal{F}^{-1}(\mathcal{F}(X)\mathcal{S}_{0:K}) = A_{K:0}XW_{0:K}$ with $\mathcal{S}_{0:K} = \prod_{i=0}^{K} \mathcal{S}_i, A_{K:0} = \prod_{i=K}^{0} A_i$ and $W_{0:K} = \prod_{i=0}^{K} W_i$. Proved. $\square$

Thus, the FourierGNN can be rewritten as (for convenience, we exclude the non-linear activation function $\sigma$ and learnable bias parameters $b$):

$$\mathcal{F}^{-1}(\sum_{k=0}^{K} \mathcal{F}(X)\mathcal{S}_{0:K}) = A_0XW_0 + A_1(A_0XW_0)W_1 + ... + A_{K:0}XW_{0:K} \tag{11}$$

From the right part of the above equation, we can observe that it assigns different weights to weigh the information of different neighbors in each diffusion order. This property enable FourierGNN to capture the complex correlations (i.e., spatiotemporal dependencies) in the hypervariate graph, which is empirically verified in our visualization experiments.

## D Compared with Other Graph Neural Networks

**Graph Convolutional Networks**. Graph convolutional networks (GCNs) depend on the Laplacian eigenbasis to perform the multi-order graph convolutions over a given graph structure. Compared with GCNs, FourierGNN as an efficient alternative to multi-order graph convolutions has three main differences: 1) No eigendecompositions or similar costly matrix operations are required. FourierGNN transforms the input into Fourier domain by discrete Fourier transform (DFT) instead of graph Fourier transform (GFT); 2) Explicitly assigning various importance to nodes of the same neighborhood with different diffusion steps. FourierGNN adopts different Fourier Graph Operators $\mathcal{S}$ in different diffusion steps corresponding to different dependencies among nodes; 3) FourierGNN is invariant to the discretization $N, T$. It parameterizes the graph convolution via Fourier Graph Operators which are invariant to the graph structure and graph scale.

**Graph Attention Networks**. Graph attention networks (GATs) are non-spectral attention-based graph neural networks. GATs use node representations to calculate the attention weights (i.e., edge weights) varying with different graph attention layers. Accordingly, both GATs and FourierGNN do

not depend on eigendecompositions and adopt varying edge weights with different diffusion steps (layers). However, FourierGNN can efficiently perform graph convolutions in Fourier space. For a complete graph, the time complexity of the attention calculation of $K$ layers is proportional to $Kn^2$ where $n$ is the number of nodes, while a $K$-layer FourierGNN infers the graph structure in Fourier space with the time complexity proportional to $n \log n$. In addition, compared with GATs that implicitly achieve edge-varying weights with different layers, FourierGNN adopts different FGOs in different diffusion steps explicitly.

# E  Experiment Details

## E.1  Datasets

We use seven public multivariate benchmarks for multivariate time series forecasting and these benchmark datasets are summarized in Table 7.

Table 7: Summary of datasets.

| Datasets | Solar | Wiki | Traffic | ECG | Electricity | COVID-19 | METR-LA |
|---|---|---|---|---|---|---|---|
| Samples | 3650 | 803 | 10560 | 5000 | 140211 | 335 | 34272 |
| Variables | 592 | 2000 | 963 | 140 | 370 | 55 | 207 |
| Granularity | 1hour | 1day | 1hour | - | 15min | 1day | 5min |
| Start time | 01/01/2006 | 01/07/2015 | 01/01/2015 | - | 01/01/2011 | 01/02/2020 | 01/03/2012 |

**Solar**[2]: This dataset is about solar power collected by National Renewable Energy Laboratory. We choose the power plant data points in Florida as the data set which contains 593 points. The data is collected from 2006/01/01 to 2016/12/31 with the sampling interval of every 1 hour.

**Wiki [34]**: This dataset contains a number of daily views of different Wikipedia articles and is collected from 2015/7/1 to 2016/12/31. It consists of approximately $145k$ time series and we randomly choose $2k$ from them as our experimental data set.

**Traffic [34]**: This dataset contains hourly traffic data from 963 San Francisco freeway car lanes. The traffic data are collected since 2015/01/01 with the sampling interval of every 1 hour.

**ECG**[3]: This dataset is about Electrocardiogram(ECG) from the UCR time-series classification archive [38]. It contains 140 nodes and each node has a length of 5000.

**Electricity**[4]: This dataset contains the electricity consumption of 370 clients and is collected since 2011/01/01. The data sampling interval is every 15 minutes.

**COVID-19**[5]: This dataset is about COVID-19 hospitalization in the U.S. states of California (CA) from 01/02/2020 to 31/12/2020 provided by the Johns Hopkins University with the sampling interval of every one day.

**METR-LA**[6]: This dataset contains traffic information collected from loop detectors in the highway of Los Angeles County from 01/03/2012 to 30/06/2012. It contains 207 sensors and the data sampling interval is every 5 minutes.

## E.2  Baselines

In experiments, we conduct a comprehensive comparison of the forecasting performance between our FourierGNN and representative and state-of-the-art (SOTA) models as follows.

**VAR** [32]: VAR is a classic linear autoregressive model. We use the Statsmodels library (`https://www.statsmodels.org`) which is a Python package that provides statistical computations to realize the VAR.

---

[2]`https://www.nrel.gov/grid/solar-power-data.html`

[3]`http://www.timeseriesclassification.com/description.php?Dataset=ECG5000`

[4]`https://archive.ics.uci.edu/ml/datasets/ElectricityLoadDiagrams20112014`

[5]`https://github.com/CSSEGISandData/COVID-19`

[6]`https://github.com/liyaguang/DCRNN`

**DeepGLO** [34]: DeepGLO models the relationships among variables by matrix factorization and employs a temporal convolution neural network to introduce non-linear relationships. We download the source code from: `https://github.com/rajatsen91/deepglo`. We follow the recommended configuration as our experimental settings for wiki, electricity, and traffic datasets. For covid datasets, the vertical and horizontal batch size is set to 64, the rank of the global model is set to 64, the number of channels is set to [32, 32, 32, 1], and the period is set to 7.

**LSTNet** [33]: LSTNet uses a CNN to capture inter-variable relationships and an RNN to discover long-term patterns. We download the source code from: `https://github.com/laiguokun/LSTNet`. In our experiment, we use the recommended configuration where the number of CNN hidden units is 100, the kernel size of the CNN layers is 4, the dropout is 0.2, the RNN hidden units is 100, the number of RNN hidden layers is 1, the learning rate is 0.001 and the optimizer is Adam.

**TCN** [8]: TCN is a causal convolution model for regression prediction. We download the source code from: `https://github.com/locuslab/TCN`. We utilize the same configuration as the polyphonic music task exampled in the open source code where the dropout is 0.25, the kernel size is 5, the number of hidden units is 150, the number of levels is 4 and the optimizer is Adam.

**Reformer** [35]: Reformer combines the modeling capacity of a Transformer with an architecture that can be executed efficiently on long sequences and with small memory use. We download the source code from: `https://github.com/thuml/Autoformer`. We follow the recommended configuration as the experimental settings.

**Informer** [9]: Informer leverages an efficient self-attention mechanism to encode the dependencies among variables. We download the source code from: `https://github.com/zhouhaoyi/Informer2020`. We follow the recommended configuration as our experimental settings where the dropout is 0.05, the number of encoder layers is 2, the number of decoder layers is 1, the learning rate is 0.0001, and the optimizer is Adam.

**Autoformer** [4]: Autoformer proposes a decomposition architecture by embedding the series decomposition block as an inner operator, which can progressively aggregate the long-term trend part from intermediate prediction. We download the source code from: `https://github.com/thuml/Autoformer`. We follow the recommended configuration as our experimental settings with 2 encoder layers and 1 decoder layer.

**FEDformer** [27]: FEDformer proposes an attention mechanism with low-rank approximation in frequency and a mixture of expert decomposition to control the distribution shifting. We download the source code from: `https://github.com/MAZiqing/FEDformer`. We use FEB-f as the Frequency Enhanced Block and select the random mode with 64 as the experimental mode.

**SFM** [24]: On the basis of the LSTM model, SFM introduces a series of different frequency components in the cell states. We download the source code from: `https://github.com/z331565360/State-Frequency-Memory-stock-prediction`. We follow the recommended settings where the learning rate is 0.01, the frequency dimension is 10, the hidden dimension is 10 and the optimizer is RMSProp.

**StemGNN** [10]: StemGNN leverages GFT and DFT to capture dependencies among variables in the frequency domain. We download the source code from: `https://github.com/microsoft/StemGNN`. We use the recommended configuration of stemGNN as our experiment setting where the optimizer is RMSProp, the learning rate is 0.0001, the number of stacked layers is 5, and the dropout rate is 0.5.

**MTGNN** [13]: MTGNN proposes an effective method to exploit the inherent dependency relationships among multiple time series. We download the source code from: `https://github.com/nnzhan/MTGNN`. Because the experimental datasets have no static features, we set the parameter load_static_feature to false. We construct the graph by the adaptive adjacency matrix and add the graph convolution layer. Regarding other parameters, we adopt the recommended settings.

**GraphWaveNet** [17]: GraphWaveNet introduces an adaptive dependency matrix learning to capture the hidden spatial dependency. We download the source code from: `https://github.com/nnzhan/Graph-WaveNet`. Since our datasets have no prior defined graph structures, we use only adaptive adjacent matrix. We add a graph convolution layer and randomly initialize the adjacent matrix. We adopt the recommended configuration as our experimental settings where the learning rate is 0.001, the dropout is 0.3, the number of epochs is 50, and the optimizer is Adam.

**AGCRN** [2]: AGCRN proposes a data-adaptive graph generation module for discovering spatial correlations from data. We download the source code from: `https://github.com/LeiBAI/AGCRN`. We follow the recommended configuration as our experimental settings where the embedding dimension is 10, the learning rate is 0.003, and the optimizer is Adam.

**TAMP-S2GCNets** [11]: TAMP-S2GCNets explores the utility of MP to enhance knowledge representation mechanisms within the time-aware DL paradigm. We download the source code from: `https://www.dropbox.com/sh/n0ajd5l0tdeyb80/AABGn-ejfV1YtRwjf_LOAOsNa?dl=0`. TAMP-S2GCNets requires predefined graph topology and we use the California State topology provided by the source code as input. We adopt the recommended configuration as our experimental settings on COVID-19.

**DCRNN** [18]: DCRNN uses bidirectional graph random walk to model spatial dependency and recurrent neural network to capture the temporal dynamics. We download the source code from: `https://github.com/liyaguang/DCRNN`. We follow the recommended configuration as our experimental settings with the batch size is 64, the learning rate is 0.01, the input dimension is 2 and the optimizer is Adam. DCRNN requires a pre-defined graph structure and we use the adjacency matrix as the pre-defined structure provided by the METR-LA dataset.

**STGCN** [1]: STGCN integrates graph convolution and gated temporal convolution through spatial-temporal convolutional blocks. We download the source code from:`https://github.com/VeritasYin/STGCN_IJCAI-18`. We use the recommended configuration as our experimental settings where the batch size is 50, the learning rate is 0.001 and the optimizer is Adam. STGCN requires a pre-defined graph structure and we leverage the adjacency matrix as the pre-defined structures provided by the METR-LA dataset.

**CoST** [36]: CoST separates the representation learning and downstream forecasting task and proposes a contrastive learning framework that learns disentangled season-trend representations for time series forecasting tasks. We download the source code from: `https://github.com/salesforce/CoST`. We set the representation dimension to 320, the learning rate to 0.001, and the batch size to 32. Inputs are min-max normalization, we perform a 70/20/10 train/validation/test split for the METR-LA dataset and 60/20/20 for the COVID-19 dataset.

## E.3 Evaluation Metrics

We use MAE (Mean Absolute Error), RMSE (Root Mean Square Error), and MAPE (Mean Absolute Percentage Error) as the evaluation metrics in the experiments.

Specifically, given the groudtruth at timestamps $t$, $Y_t = [\boldsymbol{x}_{t+1}, ..., \boldsymbol{x}_{t+\tau}] \in \mathbb{R}^{N \times \tau}$, and the predictions $\hat{Y}_t = [\hat{\boldsymbol{x}}_{t+1}, ..., \hat{\boldsymbol{x}}_{t+\tau}] \in \mathbb{R}^{N \times \tau}$ for future $\tau$ steps at timestamp $t$, the metrics are defined as follows:

$$MAE = \frac{1}{\tau N} \sum_{i=1}^{N} \sum_{j=1}^{\tau} |x_{ij} - \hat{x}_{ij}| \tag{12}$$

$$RMSE = \sqrt{\frac{1}{\tau N} \sum_{i=1}^{N} \sum_{j=1}^{\tau} (x_{ij} - \hat{x}_{ij})^2} \tag{13}$$

$$MAPE = \frac{1}{\tau N} \sum_{i=1}^{N} \sum_{j=1}^{\tau} \left| \frac{x_{ij} - \hat{x}_{ij}}{x_{ij}} \right| \times 100\% \tag{14}$$

with $x_{ij} \in Y_t$ and $\hat{x}_{ij} \in \hat{Y}_t$.

## E.4 Experimental Settings

We summarize the implementation details of the proposed FourierGNN as follows. Note that the details of the baselines are introduced in their corresponding descriptions (see Section E.2).

**Network details.** The fully connected feed-forward network (FFN) consists of three linear transformations with $LeakyReLU$ activations in between. The FFN is formulated as follows:

$$\mathbf{X}_1 = \text{LeakyReLU}(\mathbf{Y}_t^{\mathcal{G}}\mathbf{W}_1 + \mathbf{b}_1)$$
$$\mathbf{X}_2 = \text{LeakyReLU}(\mathbf{X}_1\mathbf{W}_2 + \mathbf{b}_2) \quad (15)$$
$$\hat{Y} = \mathbf{X}_2\mathbf{W}_3 + \mathbf{b}_3$$

where $\mathbf{W}_1 \in \mathbb{R}^{(Td) \times d_1^{ffn}}$, $\mathbf{W}_2 \in \mathbb{R}^{d_1^{ffn} \times d_2^{ffn}}$ and $\mathbf{W}_3 \in \mathbb{R}^{d_2^{ffn} \times \tau}$ are the weights of the three layers respectively, and $\mathbf{b}_1 \in \mathbb{R}^{d_1^{ffn}}$, $\mathbf{b}_2 \in \mathbb{R}^{d_2^{ffn}}$ and $\mathbf{b}_3 \in \mathbb{R}^{\tau}$ are the biases of the three layers respectively. Here, $d_1^{ffn}$ and $d_2^{ffn}$ are the dimensions of the three layers. In addition, we adopt a $ReLU$ activation function in Equation 6.

**Training details.** We carefully tune the hyperparameters, including the embedding size, batch size, $d_1^{ffn}$ and $d_2^{ffn}$, on the validation set and choose the settings with the best performance for FourierGNN on different datasets. Specifically, the embedding size and batch size are tuned over $\{32, 64, 128, 256, 512\}$ and $\{2, 4, 8, 16, 32, 64, 128\}$ respectively. For the COVID-19 dataset, the embedding size is 256, and the batch size is set to 4. For the Traffic, Solar, and Wiki datasets, the embedding size is 128, and the batch size is set to 2. For the METR-LA, ECG, and Electricity datasets, the embedding size is 128, and the batch size is set to 32.

To reduce the number of parameters, we adopt a linear transform to reshape the original time domain representation $\mathbf{Y}_t^{\mathcal{G}} \in \mathbb{R}^{NT \times d}$ to $\mathbf{Y}_t \in \mathbb{R}^{N \times T \times d}$, and map $\mathbf{Y}_t$ to a low-dimensional tensor $\mathbf{Y}_t \in \mathbb{R}^{N \times l \times d}$ with $l < T$. We then reshape $\mathbf{Y}_t \in \mathbb{R}^{N \times (ld)}$ and feed it to FFN. We perform a grid search on the dimensions of FFN, i.e., $d_1^{ffn}$ and $d_2^{ffn}$, over $\{32, 64, 128, 256, 512\}$ and tune the intermediate dimension $l$ over $\{2, 4, 6, 8, 12\}$. The settings of the three hyperparameters over all datasets are shown in Table 8. By default, we set the diffusion step (layers) $K = 3$ for all datasets.

Table 8: Dimension settings of FFN on different datasets. $*$ denotes that we feed the original time domain representation to FFN without the dimension reduction.

| Datasets | Solar | Wiki | Traffic | ECG | Electricity | COVID-19 | META-LR |
|---|---|---|---|---|---|---|---|
| $l$ | 6 | 2 | 2 | $*$ | 4 | 8 | 4 |
| $d_1^{ffn}$ | 64 | 64 | 64 | 64 | 64 | 256 | 64 |
| $d_2^{ffn}$ | 256 | 256 | 256 | 256 | 256 | 512 | 256 |

### E.5 Details for Visualization Experiments

To verify the effectiveness of FourierGNN in learning the spatiotemporal dependencies on the hypervariate graph, we obtain the output of FourierGNN as the node representation, denoted as $\mathbf{Y}_t^{\mathcal{G}} = \text{IDFT}(\text{FourierGNN}(\mathbf{X}_t^{\mathcal{G}})) \in \mathbb{R}^{NT \times d}$ with Inverse Discrete Fourier Transform (IDFT). Then, we visualize the adjacency matrix $\mathbf{A}$ calculated based the flatten node representation $\mathbf{Y}_t^{\mathcal{G}} \in \mathbb{R}^{NT \times d}$, formulated as $\mathbf{A} = \mathbf{Y}_t^{\mathcal{G}}(\mathbf{Y}_t^{\mathcal{G}})^T \in \mathbb{R}^{NT \times NT}$, to show the variable correlations. Note that $\mathbf{A}$ is normalized via $\mathbf{A}/\max(\mathbf{A})$. Since it is not feasible to directly visualize the huge adjacency matrix $\mathbf{A}$ of the hypervariate graph, we visualize its different subgraphs in Figures 3, 4, 9, and 10 to better verify the learned spatiotemporal information on the hypervariate graph from different perspectives.

Figure 3. We select 8 counties and visualize the correlations between 12 consecutive time steps for each selected county respectively. Figure 3 reflects the temporal correlations within each variable.

Figure 4: On the METR-LA dataset, we average its adjacency matrix $\mathbf{A}$ over the temporal dimension (i.e., marginalizing $T$) to $\mathbf{A}' \in \mathbb{R}^{N \times N}$. Then, we randomly select 20 detectors out of all $N = 207$ detectors and obtain their corresponding sub adjacency matrix ($\mathbb{R}^{20 \times 20}$) from $\mathbf{A}'$ for visualization. We further compare the sub-adjacency with the real road map (generated by the Google map tool) to verify the learned dependencies between different detectors.

Figure 9. Since we adopt a 3-layer FourierGNN, we can calculate four adjacency matrices based on the spectrum input $\mathcal{X}_t^{\mathcal{G}}$ of FourierGNN and the outputs of each layer in FourierGNN. Following the

way of visualization in Figure 4, we select 10 counties and two timestamps on the four adjacency matrices for visualization. Figure 9 shows the effects of each layer of FourierGNN in filtering or enhancing variable correlations.

Figure 10. On the COVID-19 dataset, we randomly select 10 counties out of $N = 55$ counties and obtain their four sub-adjacency matrices of four consecutive days for visualization. Each of the four sub adjacency matrices $\mathbb{R}^{10 \times 10}$ embodies the dependencies between counties in one day. Figure 10 reflects the time-varying dependencies between counties (i.e., variables).

## F Additional Results

To further evaluate the performance of our model FourierGNN in multi-step forecasting, we conduct more experiments on the Wiki, METR-LA, and ECG datasets, respectively. We compare our model FourierGNN with five models (including StemGNN [10], AGCRN [2], GraphWaveNet [17], MTGNN [13], and Informer [9]) on the Wiki dataset under different prediction lengths, and the results are shown in Table 9. From the table, we observe that FourierGNN outperforms other models on MAE, RMSE, and MAPE metrics for all the prediction lengths. On average, FourierGNN improves MAE, RMSE, and MAPE by 7.4%, 3.5%, and 22.3%, respectively. Among these models, AGCRN shows promising performances since it captures the spatial and temporal correlations adaptively. However, it fails to simultaneously capture spatiotemporal dependencies, limiting its forecasting performance. In contrast, our model captures comprehensive spatiotemporal dependencies simultaneously on a hypervariate graph for multivariate time series forecasting.

Table 9: Accuracy comparison under different prediction lengths on the Wiki dataset.

| Length | 3 | | | 6 | | | 9 | | | 12 | | |
|--------|-----|------|---------|-----|------|---------|-----|------|---------|-----|------|---------|
| Metrics | MAE | RMSE | MAPE(%) | MAE | RMSE | MAPE(%) | MAE | RMSE | MAPE(%) | MAE | RMSE | MAPE(%) |
| GraphWaveNet [17] | 0.061 | 0.105 | 138.60 | 0.061 | 0.105 | 135.32 | 0.061 | 0.105 | 132.52 | 0.061 | 0.104 | 136.12 |
| StemGNN [10] | 0.157 | 0.236 | 89.00 | 0.159 | 0.233 | 98.01 | 0.232 | 0.311 | 142.14 | 0.220 | 0.306 | 125.40 |
| AGCRN [2] | 0.043 | 0.077 | 73.49 | 0.044 | 0.078 | 80.44 | 0.045 | 0.079 | 81.89 | 0.044 | 0.079 | 78.52 |
| MTGNN [13] | 0.102 | 0.141 | 123.15 | 0.091 | 0.133 | 91.75 | 0.074 | 0.120 | 85.44 | 0.101 | 0.140 | 122.96 |
| Informer [9] | 0.053 | 0.089 | 85.31 | 0.054 | 0.090 | 84.46 | 0.059 | 0.095 | 93.80 | 0.059 | 0.095 | 95.09 |
| **FourierGNN** | **0.040** | **0.075** | **58.18** | **0.041** | **0.075** | **60.43** | **0.041** | **0.076** | **60.95** | **0.041** | **0.076** | **64.50** |

Table 10: Accuracy comparison under different prediction lengths on the METR-LA dataset.

| Horizon | 3 | | | 6 | | | 9 | | | 12 | | |
|---------|-----|------|---------|-----|------|---------|-----|------|---------|-----|------|---------|
| Metrics | MAE | RMSE | MAPE(%) | MAE | RMSE | MAPE(%) | MAE | RMSE | MAPE(%) | MAE | RMSE | MAPE(%) |
| DCRNN [18] | 0.160 | 0.204 | 80.00 | 0.191 | 0.243 | 83.15 | 0.216 | 0.269 | 85.72 | 0.241 | 0.291 | 88.25 |
| STGCN [1] | 0.058 | 0.133 | 59.02 | 0.080 | 0.177 | 60.67 | 0.102 | 0.209 | 62.08 | 0.128 | 0.238 | 63.81 |
| GraphWaveNet [17] | 0.180 | 0.366 | 21.90 | 0.184 | 0.375 | 22.95 | 0.196 | 0.382 | 23.61 | 0.202 | 0.386 | 24.14 |
| MTGNN [13] | 0.135 | 0.294 | **17.99** | 0.144 | 0.307 | **18.82** | 0.149 | 0.328 | **19.38** | 0.153 | 0.316 | **19.92** |
| StemGNN [10] | 0.052 | 0.115 | 86.39 | 0.069 | 0.141 | 87.71 | 0.080 | 0.162 | 89.00 | 0.093 | 0.175 | 90.25 |
| AGCRN [2] | 0.062 | 0.131 | 24.96 | 0.086 | 0.165 | 27.62 | 0.099 | 0.188 | 29.72 | 0.109 | 0.204 | 31.73 |
| Informer [9] | 0.076 | 0.141 | 69.96 | 0.088 | 0.163 | 70.94 | 0.096 | 0.178 | 72.26 | 0.100 | 0.190 | 72.54 |
| CoST [36] | 0.064 | 0.118 | 88.44 | 0.077 | 0.141 | 89.63 | 0.088 | **0.159** | 90.56 | 0.097 | 0.171 | 91.42 |
| **FourierGNN** | **0.050** | **0.113** | 86.30 | **0.066** | **0.140** | 87.97 | **0.076** | **0.159** | 88.99 | **0.084** | **0.165** | 89.69 |

Furthermore, we compare our model FourierGNN with seven MTS models (including STGCN [1], DCRNN [18], StemGNN [10], AGCRN [2], GraphWaveNet [17], MTGNN [13], Informer [9], and CoST [36]) on the METR-LA dataset which has a predefined graph topology in the data, and the results are shown in Table 10. On average, we improve 5.7% on MAE and 1.5% on RMSE. Among these models, StemGNN achieves competitive performance because it combines GFT to capture the spatial dependencies and DFT to capture the temporal dependencies. However, it is also limited to simultaneously capturing spatiotemporal dependencies. CoST learns disentangled seasonal-trend representations for time series forecasting via contrastive learning and obtains competitive results. But, our model still outperforms CoST. Because, compared with CoST, our model not only can learn the dynamic temporal representations, but also capture the discriminative spatial representations. Besides, STGCN and DCRNN require pre-defined graph structures. But StemGNN and our model outperform them for all steps, and AGCRN outperforms them when the prediction lengths are 9 and 12. This also shows that a novel adaptive graph learning can precisely capture the hidden spatial dependency. In addition, we compare FourierGNN with the baseline models under the different prediction lengths on the ECG dataset, as shown in Figure 5. It reports that FourierGNN achieves the best performances (MAE, RMSE, and MAPE) for all prediction lengths.

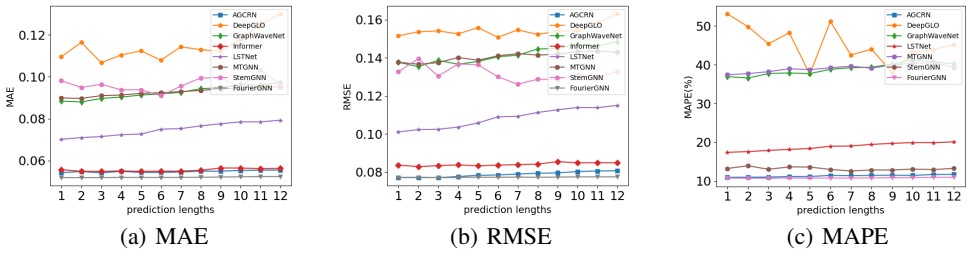

(a) MAE     (b) RMSE     (c) MAPE

Figure 5: Performance comparison in multi-step prediction on the ECG dataset.

# G Further Analyses

## G.1 Scalability Analysis

We further conduct experiments on the Wiki dataset to investigate the performance of FourierGNN under different graph sizes ($N \times T$). The results are shown in Figure 6, where Figure 6(a), Figure 6(b) and Figure 6(c) show MAE, RMSE, and MAPE at the different number of nodes, respectively. From these figures, we observe that FourierGNN keeps a leading edge over the other state-of-the-art MTS models as the number of nodes increases. The results demonstrate the superiority and scalability of FourierGNN on large-scale datasets.

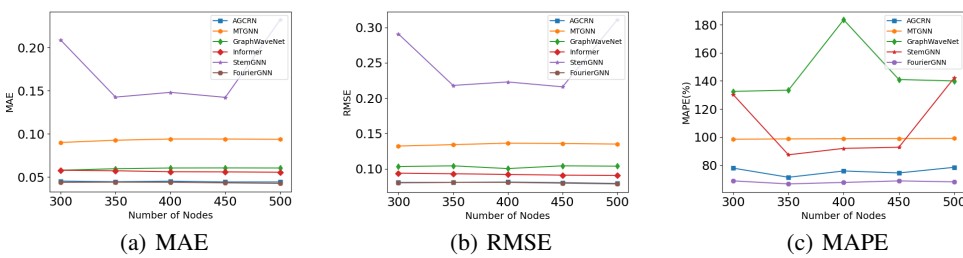

(a) MAE     (b) RMSE     (c) MAPE

Figure 6: Scalability analyses in terms of MAE, RMSE, and MAPE under different number of nodes on the Wiki dataset.

## G.2 Parameter Analysis

We evaluate the forecasting performance of our model FourierGNN under different diffusion steps (layers) on the COVID-19 dataset, as illustrated in Table 11. The table shows that FourierGNN achieves increasingly better performance from $K = 1$ to $K = 4$ and achieves the best results when $K = 3$. With the further increase of $K$, FourierGNN obtains inferior performance. The results indicate that high-order diffusion information

Table 11: Performance at different diffusion steps (layers) on the COVID-19 dataset.

|         | K=1   | K=2   | K=3   | K=4   |
|---------|-------|-------|-------|-------|
| MAE     | 0.136 | 0.133 | 0.123 | 0.132 |
| RMSE    | 0.181 | 0.177 | 0.168 | 0.176 |
| MAPE(%) | 72.30 | 71.80 | 71.52 | 72.59 |

is beneficial for improving forecasting accuracy, but the diffusion information may gradually weaken the effect or even bring noises to forecasting with the increase of the order.

In addition, we conduct additional experiments on the ECG dataset to analyze the effect of the input lookback window length $T$ and the embedding dimension $d$, as shown in Figure 7 and Figure 8, respectively. Figure 7 shows that the performance (including RMSE and MAPE) of FourierGNN gets better as the input lookback window length increases, indicating that FourierGNN can learn a comprehensive hypervariate graph from long MTS inputs to capture the spatial and temporal dependencies. Moreover, Figure 8 shows that the performance (RMSE and MAPE) first increases and then decreases with the increase of the embedding size, which is attributed that a large embedding

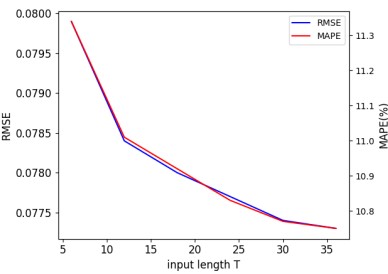
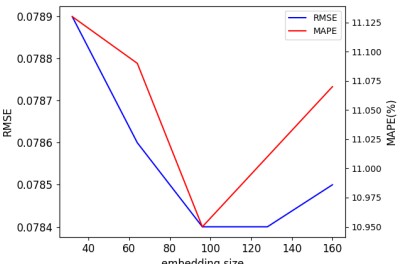

Figure 7: Influence of input window.      Figure 8: Influence of embedding size.

size improves the fitting ability of FourierGNN but it may easily lead to the overfitting issue especially when the embedding size is too large.

### G.3 Ablation Study

We provide more details about each variant used in this section and Section 5.3.

- **w/o Embedding**. A variant of FourierGNN feeds the raw MTS input instead of its embeddings into the graph convolution in Fourier space.
- **w/o Dynamic FGO**. A variant of FourierGNN uses the same FGO for all diffusion steps instead of applying different FGOs in different diffusion steps. It corresponds to a vanilla graph filter.
- **w/o Residual**. A variant of FourierGNN does not have the $K = 0$ layer output, i.e., $\mathcal{X}_t^{\mathcal{G}}$, in the summation.
- **w/o Summation**. A variant of FourierGNN adopts the last order (layer) output as the final frequency output of the FourierGNN.

We conduct another ablation study on the COVID-19 dataset to further investigate the effects of the different components of our FourierGNN. The results are shown in Table 12, which confirms the results in Table 4 and further verifies the effectiveness of each component in FourierGNN. Both Table 12 and Table 4 report that the embedding and dynamic FGOs in FourierGNN contribute more than the design of residual and summation to the state-of-the-art performance of FourierGNN.

Table 12: Ablation studies on the COVID-19 dataset.

| Metric | w/o Embedding | w/o Dynamic FGO | w/o Residual | w/o Summation | FourierGNN |
|---|---|---|---|---|---|
| MAE | 0.157 | 0.138 | 0.131 | 0.134 | 0.123 |
| RMSE | 0.203 | 0.180 | 0.174 | 0.177 | 0.168 |
| MAPE(%) | 76.91 | 74.01 | 72.25 | 72.57 | 71.52 |

## H Visualizations

### H.1 Visualization of the Diffusion Process in FourierGNN

To gain insight into the operation of the FGO, we visualize the frequency output of each layer in our FourierGNN. We select 10 counties from the COVID-19 dataset and visualize their adjacency matrices at two different timestamps, as shown in Figure 9. From left to right, the results correspond to the original spectrum of the input, as well as the outputs of the first, second, and third layers of the FourierGNN. From the top, we can find that as the number of layers increases, some correlation values are reduced, indicating that some correlations are filtered out. In contrast, the bottom case illustrates some correlations are enhanced as the number of layers increases. These results show that FGO can adaptively and effectively capture important patterns while removing noises, enabling the learning of a discriminative model.

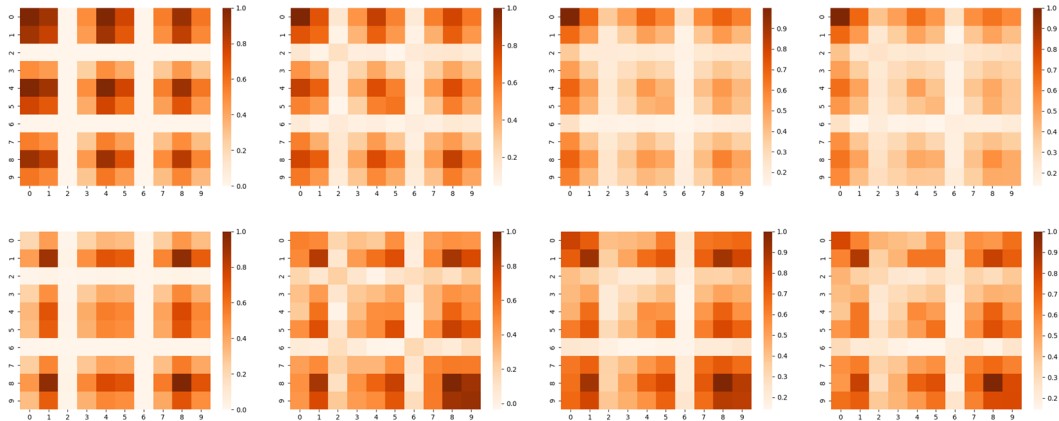

Figure 9: The diffusion process of FourierGNN at two timestamps (top and bottom) on COVID-19.

## H.2 Visualization of Time-Varying Dependencies Learned by FourierGNN

Furthermore, we explore the capability of FourierGNN in capturing time-varying dependencies among variables. To investigate this, we perform additional experiments to visualize the adjacency matrix of 10 randomly-selected counties over four consecutive days on the COVID-19 dataset. The visualization results, displayed as a heatmap in Figure 10, reveal clear spatial patterns that exhibit continuous evolution in the temporal dimension. This is because FourierGNN can attend to the time-varying variability of the spatiotemporal dependencies. These results verify that our model enjoys the feasibility of exploiting the time-varying dependencies among variables.

Based on the insights gained from these visualization results, we can conclude that the hypervariate graph structure exhibits strong capabilities to encode spatiotemporal dependencies. By incorporating FGOs, FourierGNN can effectively attend to and exploit the time-varying dependencies among variates. The synergy between the hypervariate graph structure and FGOs empowers FourierGNN to capture and model intricate spatiotemporal relationships with remarkable effectiveness.

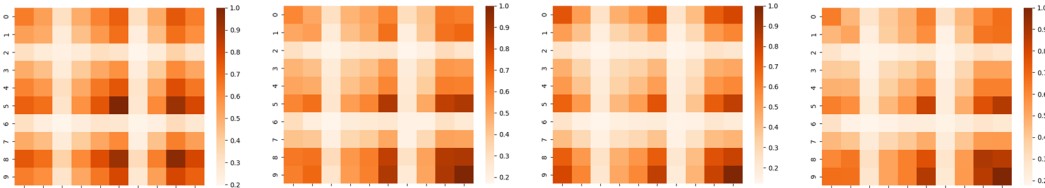

Figure 10: The adjacency matrix for four consecutive days on the COVID-19 dataset.

