# OpenReview forum: "FourierGNN: Rethinking Multivariate Time Series Forecasting from a Pure Graph Perspective"
_NeurIPS.cc/2023/Conference — NeurIPS 2023 poster_

### Official Review · Reviewer_yaJh · 2023-07-01

**Soundness:** 3 good
**Presentation:** 3 good
**Contribution:** 3 good
**Rating:** 7
**Confidence:** 4

**Summary:**

The paper presents FourierGNN for multivariate time series forecasting from a pure graph perspective, performing matrix multiplications in Fourier space, which has not been investigated so far. They design a new hypervariate graph structure to consider spatiotemporal dynamics unitedly and reformulate the graph operations on the hypervariate graph in Fourier space. Extensive experiments have demonstrated superior performance with higher efficiency and fewer parameters over state-of-the-art methods.

**Strengths:**

1. The paper is overall well-written. I think the contribution and novelty of the proposed work are clearly presented.
2. The idea of directly applying graph networks for multivariate time series forecasting seems novel and exciting. The hypervariate graph structure can encode spatiotemporal dependencies unitedly, and the visualizations in the experiments have also verified its advantages.
3. The authors argue that performing the multiplication between the input and the proposed FGO in Fourier space is equivalent to a graph convolution in the time domain, while the multiplications in Fourier space have lower complexity. The argument is proved by the authors and seems very meaningful. I think this argument provides a new path for conducting graph operations in Fourier space.
4. The effectiveness is validated through extensive experiments across seven real-world datasets. Sufficient analysis including efficiency analysis and ablation study is also provided.


**Weaknesses:**

1. In the efficiency analysis section, the authors assess the overall efficiency of the forecasting process. To specifically evaluate the efficiency of FourierGNN, it would be more appropriate to focus on comparing the number of parameters involved in the graph operations with those of the baselines.
2. The MAPE (Mean Absolute Percentage Error) results on some datasets (e.g., Traffic, COVID-19) do not achieve state-of-the-art performance.


**Questions:**

FourierGNN demonstrates impressive performance across multiple datasets, with notable success observed on the COVID dataset. Could you explain the reasons that contribute to the superior performance of FourierGNN on the COVID dataset compared to other datasets?

---

> ### Author Rebuttal · Authors · 2023-08-09
>
> Thank you for your positive feedback and valuable suggestions. In the following, we provide a detailed response to address all of your concerns.
>
> **W1**
>
> Since it is difficult and unfair to analyze the time complexity and parameter volumes of comparative methods that have different architectures, we report the empirical training time and parameter volumes of FourierGNN and the GNN-based baselines in the efficiency analysis. Meanwhile, we provide theoretical efficiency analysis between FGO/FourierGNN and graph convolution/GCN (a commonly-used module) in the analysis of time complexity.
>
> The efficiency analysis in **Model Analysis in Section 5.3** aims to investigate the parameter volumes and training time costs of FourierGNN and its comparative baselines. The comparison results show:
> 1) FourierGNN exhibits the lowest volume of parameters among the comparative models;
> 2) FourierGNN runs much faster than all baseline models.
>
> These results demonstrate the efficiency of our proposed FourierGNN compared with state-of-the-art GNN-based models.
>
> Regarding the time complexity of the graph operators in FourierGNN, we have discussed and provided the time complexity of FGO in **Complexity Analysis (Lines 203-209)**. Specifically, the computational time complexity of FGO is $\mathcal{O}(nd\operatorname{log}n+nd^2)$, while the time complexity of the equivalent graph convolution in the time domain, i.e., $AXW$, is $\mathcal{O}(n^2d+nd^2)$. This indicates the lower time complexity of FGO compared with the graph convolution. Accordingly, FourierGNN ($K$-order) has the time complexity of $\mathcal{O}(nd\operatorname{log}n+Knd^2)$, compared with the time complexity of $K$-layer GCNs $\mathcal{O}(Kn^2d+Knd^2)$. In addition, the parameter volumes of FGO and FourierGNN are $\mathcal{O}(d^2)$ and $\mathcal{O}(Kd^2)$ respectively, which are the same as those of the graph convolution and GCNs. We summarize the parameter volumes and time complexity in **Table 5 in the attached PDF**.
>
> **W2.**
>
> While FourierGNN does not achieve the best MAPE results on all datasets, it consistently ranks within the top-3 among all 14 comparative methods and demonstrates comparable performance to state-of-the-art MAPE results.
>
> Furthermore, FourierGNN outperforms all the baselines in terms of MAE and RMSE, even though it may not achieve the best MAPE scores. This is an acceptable outcome because the best baseline on different datasets does not necessarily guarantee the best performance in terms of MAE, RMSE, and MAPE. These three metrics reflect the accuracy of forecasting from different perspectives. In addition, FourierGNN, along with most of the baseline methods, adopts MSE as the objective function. As a result, the comparative methods tend to prioritize improvements in terms of MSE or RMSE.
>
> In summary, FourierGNN achieves the best MAE and RMSE scores while consistently ranks within the top-3 in terms of MAPE scores compared to state-of-the-art baselines. These results indicate the superiority of FourierGNN over other state-of-the-art models for MTS forecasting.
>
> **Q1.**
>
> Therefore, the outstanding performance of FourierGNN on the COVID-19 dataset can be attributed to its ability to effectively capture and model the complex spatiotemporal dependencies present in the data.
>
> The superior performance of FourierGNN on the COVID-19 dataset is reasonable and can be attributed to the significant spatiotemporal dependencies in the dataset. As shown in **Section E.1 Datasets**, the dataset is about COVID-19 hospitalization in the U.S. states of California (CA) from 01/02/2020 to 31/12/2020 provided by the Johns Hopkins University. The COVID-19 dataset exhibits characteristics that align well with the principles of infectious disease transmission across different regions over time. Consequently, the variables in the dataset demonstrate high correlation with each other, and the datapoints within each variable and across different variables exhibit temporal correlation. These characteristics make the COVID-19 dataset particularly suitable for evaluating the effectiveness of our proposed FourierGNN in capturing spatiotemporal dependencies.
>
> To interpret/verify the capability of FourierGNN in spatiotemporal modeling for MTS forecasting, we conducted visualizations of the learned adjacency matrix from different perspectives:
> 1) Temporal Adjacency Matrix: In **Figure 3 of Appendix H.2**, we visualize the temporal adjacency matrix of eight variables. The results clearly demonstrate that FourierGNN learns distinct temporal patterns for each variable (county), indicating that the hypervariate graph can encode rich and discriminative temporal dependencies.
>
> 2) Adjacency Matrices in Different Layers: In **Figure 9 of Appendix H.2**, we display the adjacency matrices of variables in different layers of FourierGNN. The visualization reveals that the Fourier Graph Operator (FGO) can adaptively and effectively capture important patterns while removing noise, thereby enabling the learning of a discriminative model.
>
>
> 3) Final Adjacency Matrices for Consecutive Days: In **Figure 10 of Appendix H.2**, we present the final adjacency matrices of variables for four consecutive days. These matrices highlight the time-varying dependencies among variables and showcase the feasibility of FourierGNN in exploiting such dependencies.
>
> Overall, these visualization results provide strong support for the superior performance of FourierGNN on the COVID-19 dataset, indicating its effectiveness in learning underlying node correlations and removing redundant correlations on the hypervariate graph.
>
> We'll update paper to address the above aspects and hope we have addressed your comments.

---

### Official Review · Reviewer_ipTP · 2023-07-05

**Soundness:** 4 excellent
**Presentation:** 3 good
**Contribution:** 4 excellent
**Rating:** 7
**Confidence:** 4

**Summary:**

This paper has studied a popular and important problem, i.e., modeling the intricate spatial and temporal dependencies among multivariate time series for accurate forecasting. To overcome the main limitation that existing works always separately model spatial and temporal, this work designs a new hypervariate graph structure to encode the spatiotemporal dynamics unitedly, and proposes a novel FourierGNN to learn the spatiotemporal dependencies. Extensive experiments on seven real-world datasets demonstrate FourierGNN achieves good performances in both accuracy and efficiency compared with state-of-the-art methods.

**Strengths:**

This paper presents a novel formulation from a pure graph perspective to model spatiotemporal dependencies for MTS forecasting. The work is interesting and original, which is different from previous GNN-based methods that typically model spatial and temporal dependencies with distinct graph and temporal networks. The novel pure graph modeling is straightforward yet quite meaningful for MTS forecasting, which brings up good inspiring insights.

This paper proposes a graph neural network, namely FourierGNN, to cooperate with the pure graph formulation and provides a theoretical guarantee of effectiveness.

This paper seems to have solid, extensive and diverse experiments, which are conducted on seven real-world different datasets. The extensive experimental results have clearly demonstrated the obvious performance improvement of FourierGNN in terms of accuracy and efficiency over state-of-the-art MTS forecasting models.

**Weaknesses:**

I have two concerns.
My first concern is about the hypervariate graph structure. Although the authors stated that all variables at all timestamps can encode high-resolution relationships, it also may introduce some redundant information or unwanted correlations, such as some variables that are far apart in time. How can this be properly handled? Besides, I find that the core operations of FourierGNN are conducted in the Fourier space, so I am curious whether transferring the hypervariate graph structure into the Fourier space help to learn dependencies.

In addition, another concern is whether FourierGNN can also be applied to other domains apart from the multivariate forecasting. Maybe it can be extended to other graph tasks.

Minor typos: In the caption of Figure 2, Given the hypervariate graph $\mathcal{G}=(X^\mathcal{G}_t, A^\mathcal{G}_t)$  should be $\mathcal{G}_t=(X^\mathcal{G}_t, A^\mathcal{G}_t)$

**Questions:**

1.Besides MTS forecasting, can FourierGNN be applied to other domains as well?
2.How does the proposed model handle the redundant information in the hypervariate graph structure?

---

> ### Author Rebuttal · Authors · 2023-08-09
>
> We appreciate your positive comments. We would like to respond to your comments as follows.
>
> **Q1. How to properly handled redundant or unwanted correlations in the hypervariate graph?**
>
> As stated in **Section 4.1 The Pure Graph Formulation**, we propose the hypervariate graph to connect all variables at all timestamps, where we view spatial dynamics and temporal dependencies from a united perspective. It benefits modeling the real-world spatiotemporal inter-dependencies. Specifically, since we do not have the pre-defined graph structure connecting any two variables at any two timestamps, we initialize the hypervariate graph as a fully-connected graph and learn edge weights (i.e., node spatiotemporal inter-dependencies) adaptively to training data. Accordingly, node correlations are adaptively learned by the supervision of MTS forecasting objectives. In other words, although we initially consider connections between all variables at all timestamps, significant/redundant node correlations are remained/reduced during training FourierGNN (i.e., the learning of spatiotemporal dependencies) via adaptively adjusting the correlation values.
>
> Empirically, we have visualized the learned adjacency matrices from three different perspectives:
> 1) temporal adjacency matrix of eight variates (see **Figure 3 in Section 5.4**);
> 2) adjacency matrices of variables in different layers of FourierGNN (see **Figure 9 in Appendix H.2**); and
> 3) final adjacency matrices of variables for four consecutive days (see **Figure 10 in Appendix H.2**).
>
> From these visualization results, we can obviously observe:
> 1) FourierGNN learns distinct temporal patterns for each variable (county), indicating that the hypervariate graph can encode rich and discriminative temporal dependencies (corresponding to Figure 3);
> 2) FGO can adaptively and effectively capture important patterns while removing noises, enabling the learning of a discriminative model (corresponding to Figure 9); and
> 3) FourierGNN enjoys the feasibility of exploiting the time-varying dependencies among variables (corresponding to Figure 10).
>
> Furthermore, we have conducted a visualization analysis to evaluate the effectiveness of the learned correlations in accordance with the real-world road map on the METR-LA dataset. The results provide strong evidence that the learned hypervariate graph structure can represent highly interpretable correlations, confirming the ability of FourierGNN to capture meaningful and relevant relationships among nodes. Please refer to **Figure 4 in Section 5.4** for visual representation.
>
> These empirical findings support the notion that FourierGNN is capable of effectively learning underlying node correlations and removing redundant correlations based on the hypervariate graph.
>
> **Q2. Whether transferring the hypervariate graph structure into the Fourier space help to learn dependencies.**
>
> Yes, we transfer the hypervariate graph structure into the Fourier space, which helps learn spatiotemporal dependencies from two perspectives:
> 1) **Effectiveness**: According to the convolution theorem (see **Appendix B**), the Fourier transform of a convolution of two sequences equals the pointwise product of their Fourier transforms. The theorem demonstrates that our proposed Fourier Graph Operator (FGO) is equivalent to the graph convolution operation in the time domain as shown in **Equations 4 and 5**, which provides a theoretical basis for FGO's ability to capture spatiotemporal dependencies on the hypervariate graph. See **Explanations and Proofs in Appendix C** for more details. In addition, Fourier transform offers a global view of the data, which may benefit capturing the global characteristics of the whole sequence [20]. The advantage helps FGO to effectively capture and model the global spatiotemporal dependencies in the hypervariate graph.
> 2) **Efficiency**: As shown in **Definition 1**, the hypervariate graph $\mathcal{G}\_t$ as a fully-connected graph contains $NT$ nodes, and its corresponding adjacency matrix is ${A}^{\mathcal{G}}\_t \in \mathbb{R}^{NT \times NT}$. It is extremely time-consuming to perform GCN or GAT on the hypervariate graph since the time complexity of the graph convolution and graph attention is quadratic to the number of nodes (i.e., $(NT)^2$) and proportional to the number of edges (i.e., $(NT)^2$), respectively. In contrast, the time complexity of our proposed FGO is proportional to $NT\operatorname{log}(NT)$, where the Log-linear $\mathcal{O}(n\operatorname{log}n)$ complexity makes FourierGNN much more efficient. Please refer to **Complexity Analysis in Lines 203-209** for more details.
>
> In summary, our proposed FourierGNN transfers the hypervariate graph into the Fourier space, facilitating efficiently learning effective spatiotemporal dependencies on the hypervariate graph.
>
> **Q3. "Whether FourierGNN can also be applied to other domains ..."**
>
> Yes. In **Section 4.2 FourierGNN**, the paper presents the definition and formulation of the proposed FGO and FourierGNN, which are designed based on a graph. FGO, as well as FourierGNN, can be seen as a form of global convolution or multi-order convolutions, and they are not specifically limited to the time series domain. They can be applied to various other domains where graph-based learning models are applicable. When extending the application of FourierGNN to other domains, it is crucial to ensure that the underlying graph topology in those domains satisfies the conditions of the Green's kernel.
>
> **Q4. Minor typos**
>
> We will carefully check thoroughly the paper and correct the typos in the final version.
>
> **Q5. "... can FourierGNN be applied to other domains as well?"**
>
> Please refer to our response in Q3.
>
> **Q6. "How does the proposed model handle the redundant information...?"**
>
> Please refer to our response in Q1.
>
> We will clarify the above in the final version and hope that we have addressed all your concerns.

---

### Official Review · Reviewer_ndBv · 2023-07-06

**Soundness:** 3 good
**Presentation:** 3 good
**Contribution:** 2 fair
**Rating:** 3
**Confidence:** 4

**Summary:**

The paper addresses the time series forecasting problem.
The authors propose a model that represents each scalar
observation as a node in a (fully connected) graph,
encodes the nodes and finally regresses the future
observations on all previous encoded observations
(via a fully connected layer). As encoding they propose
a discrete fourier transform followed by several node-wise
fully connected layers and finally an inverse fourier transform.
In experiments on 6 datasets they show improvements over
several baselines.


**Strengths:**

s1. interesting problem: time series forecasting
s2. interesting approach: parametrizing transformations in fourier space
s3. results showing improvements over many baselines


**Weaknesses:**

w1. a main aspect of the proposed model is not clear.
w2. missed recent related work.
w3. experimental protocol is unclear and deviates from the literature.
w4. for a key component of the proposed model, there is no ablation study
  demonstrating its impact.


**Questions:**

w1. a main aspect of the proposed model is not clear.
- the authors describe the fourier graph layer as applying the discrete
  fourier transform on the graph features X\in\R^{nd\times 1} of a complete graph,
  explicitly not the graph fourier transform. Which dimension is the time
  dimension for this fourier transform? The node dimension of X contains
  nd many entries, one for each time point and channel. While one can
  compute a fourier transform also over this vector, what is it semantics?
  How does it capture time varying information? And would this not crucially
  depend on how on is ordering these nd many entries?

w2. missed recent related work.
- esp. PatchTST [Nie et al. 2023] and dlinear [Zeng et al. 2023] are well
  known model outperforming your baselines by a good margin.

w3. experimental protocol is unclear and deviates from the literature.
- what is the forecasting horizon \tau you are using?
- the experiments report only a single number per dataset and error measure,
  while in the literature usually  results for a portfolio of forecasting horizons
  are reported (e.g., 96, 192, 336, 720, e.g., the Fedformer). Why do you deviate
  from this standard? This way it also is not possible to compare your results
  with the results published in those papers.
- usually results are also reported for further datasets such as ETTm2, Exchange,
  Weather and ILI. How does your model compare to those published results?

w4. for a key component of the proposed model, there is no ablation study
  demonstrating its impact.
- while there are many papers that simply operate on the fourier spectrum
  of a time series, one key component of the model of the authors seems
  to be that after these operations they move back into the time domain
  by the inverse fourier transform. What is the impact of this back transformation?

references:
- Nie, Yuqi, Nam H. Nguyen, Phanwadee Sinthong, and Jayant Kalagnanam. “A Time Series Is Worth 64 Words: Long-Term Forecasting with Transformers.” arXiv, March 5, 2023. https://doi.org/10.48550/arXiv.2211.14730.
- Zeng, Ailing, Muxi Chen, Lei Zhang, and Qiang Xu. “Are Transformers Effective for Time Series Forecasting?” In AAAI, 2023.

---

> ### Author Rebuttal · Authors · 2023-08-09
>
> We appreciate your review. Hope our response can address the misunderstandings or concerns.
>
> **w1**
>
> 1. The node features of the hypervariate graph are $X \in \mathbb{R}^{n \times d}$, where $n=NT$ is the number of nodes, $d$ is the number of features, $N$ is the number of variables, and $T$ is the input length. In our proposed FourierGNN, we conduct DFT along the spatiotemporal dimension of $n$ (**see lines 227-228**).
>
>     Note that the Fourier transform allows us to transform data from the time domain to the frequency domain, revealing its frequency spectrum. However, the Fourier transform is not limited to the time domain alone; it can be applied to various types of data beyond time series, including images and other multidimensional data. Accordingly, the Fourier transform is not necessarily performed along the time dimension, for example, one can apply DFT on images to obtain the frequency spectrum features.
>
> 2. According to the convolution theorem (see **Appendix B**), the Fourier transform of a convolution of two sequences equals the pointwise product of their Fourier transforms. Therefore, the multiplication between $\mathcal{F}(X)$ and FGO $\mathcal{S}$ can be written as $\mathcal{F}(\sum_{j=1}^n {{X}}[j]\kappa[i-j])=\mathcal{F}((X*\kappa)[i])=\mathcal{F}({X})\mathcal{S}$, corresponding to a graph convolution on the hypervariate graph (see **Equations 4 and 5**). In other words, FGO is equivalent to the graph convolution, and FourierGNN is equivalent to multi-order convolutions on the hypervariate graph (**see Proposition 1**).
>
>     Mathematically, conducting DFT on the spatiotemporal dimension of $n$ is to purposefully transform the time-consuming graph convolutions on the hypervariable graph to the efficient pointwise multiplication in the Fourier space (**see Complexity Analysis in Section 4.2**). Intuitively, it obtains the global frequency spectrum of the node features of the hypervariate graph, corresponding to the values of all variables at each timestamp, facilitating learning a high-resolution spatiotemporal representation across timestamps and variables (**explanations can be seen in Appendix C.1**).
>
>     In addition, we have conducted visualization experiments to demonstrate that FourierGNN can learn both discriminative temporal dependencies and highly interpretative spatial dependencies (**Figures 3 and 4 in Section 5.4** and **Figures 9 and 10 in Appendix H Visualizations**).
>
> 3. Structure: According to **Definition 1**, the hypervariate graph connects any two variables at any two timestamps. It embodies not only the intra-series temporal dependencies (node connections of each individual variable), time-varying inter-series spatial dependencies (node connections over each single time step), and also the time-varying spatiotemporal dependencies (node connections between different variables at different time steps). More details can be seen in **Appendix C.1**.
>
>     Methodology: FourierGNN stacking multiple FGOs is equivalent to multi-order graph convolutions, enabling FourierGNN to adaptively capture the abovementioned high-resolution spatiotemporal dependencies, including the time-varying correlations.
>
>     Empirically: In **Figure 10 in Appendix H.2**, we visualize the learned adjacency matrix of 10 randomly-selected counties over four consecutive days on the COVID-19 dataset. The results reveal clear spatial patterns that exhibit continuous evolution in the temporal dimension, verifying that FourierGNN enjoys the feasibility of exploiting the time-varying dependencies among variables.
>
> 4. Our proposed FourierGNN **is not** influenced by the order of $n$ node features. Structurally, since the hypervariate graph is a fully-connected graph, the order of its $n$ nodes is trivial. Mathematically, given data $x[n]$ with $N$ datapoints, the DFT of $x$ is $\mathcal{X}[k]=\sum_{n=0}^{N-1} x[n]\cdot e^{-\frac{i2\pi }{N} kn } $. The frequency spectrum $\mathcal{X}[k]$ is regardless of the order of the datapoints $x[n]$.
>
>     To verify the claim, we randomly shuffled the order of time series variables in the raw ECG data five times and evaluated our model on each shuffled set of data. The result is reported in **Table 3 in the attached PDF**, which shows that FourierGNN achieves consistent performance on raw data and randomly shuffled data.
>
> **w2**
>
> Note that FourierGNN represents a **GNN-based** model that incorporates **frequency analysis** and is tailored for short-term multivariate time series forecasting. Accordingly, we chose GNN-based models (AGCRN, StemGNN, MTGNN, GraphWaveNet, TAMP-S2GCNets, DCRNN, and STGCN), frequency-based models (SFM, FEDformer, Autoformer, and CoST), and short-term models (LSTNet, DeepGLO, and TCN), as well as one representative model (Informer) as our baselines.
>
> PatchTST (Transformer-based) and DLinear (MLP-based) are the latest representative work for long-term time series forecasting. Considering that they are neither short-term models nor GNN-based/frequency-based models, they were not included in the comparison. **Moreover, PatchTST and DLinear have not compared their performance with GNN-based models, and they have also not conducted experiments according to the short-term settings**.
>
> To address your concern, we performed experiments to compare FourierGNN with PatchTST and DLinear on seven real-world datasets for short-term forecasting (the input length and the prediction length are 12). We reported the results in **Table 1 in the attached PDF**. From the results, we can find that FourierGNN outperforms PatchTST and DLinear on all datasets. The results are reasonable because, compared with FourierGNN, DLinear and PatchTST are more effective at handling gradually evolving trends/long-term temporal correlations but less effective at capturing complex/time-varying spatiotemporal dependencies.
>
> For W3 and W4, due to space limit, we response them in general response and PDF file.

---

> > ### Author Response · Authors · 2023-08-17
> >
> > Dear Reviewer ndBv,
> >
> > We thank you again for your review and effort. We were kindly wondering if our responses have addressed your concerns. Besides, your feedbacks are really important to us, and we are also looking forward to further discussions with you.
> >
> > Authors

---

> > ### Comment · Reviewer_ndBv · 2023-08-18
> > **answer to rebuttal**
> >
> > Dear authors, thanks for your extensive answers and additional
> > experiments. My concerns w1 and w4 you resolved. About the other
> > two:
> >
> > w2. PatchTST and dlinear
> > - The numbers you report for e.g. PatchTST deviate from the published
> >   numbers, e.g., for dataset weather, horizon 96, MAE
> >   - you report 0.034  (table 2 in your rebuttal pdf),
> >   - the PatchTST paper reports 0.198.
> >   Can you explain these differences? It would be more convincing to
> >   reproduce the experimental settings of the baseline papers (and
> >   their numbers).
> >
> > w3. experimental protocol unclear and deviates from the literature.
> > 1. for the main experiment in tab. 1: what is \tau? 1 ?
> > 2. what does "follow the experimental settings in short-term forecasting
> >   baselines, like LSTNet and StemGNN" mean exactly?
> >   - The LSTNet paper reports two different error measures, RSE and corr,
> >     so one cannot compare numbers directly.
> >   - The StemGNN paper reports RMSE for datasets Solar and Electricity,
> >     but they report different numbers than you do (their table 2):
> > 	- Solar: they report 0.07, you report 0.222
> > 	- Electricity: they report 0.06, you report 0.101
> >     Are you using a vastly different split? Or how are these differences
> >     explained?
> > 3. Do any of the numbers in your table 1 coincide with some published
> >   numbers in the baseline papers? And if so, could you mark them, say with
> >   a star?

---

> > > ### Author Response · Authors · 2023-08-19
> > > **Thanks for your feedback (1/2)**
> > >
> > > Dear Reviewer ndBv,
> > >
> > > We greatly appreciate your feedback, and thanks for your careful readings.
> > > We would like to clarify our experimental settings and the two points you mentioned.
> > >
> > > **About experimental settings**
> > >
> > > In the literature on short-term forecasting, previous models have employed diversified experimental settings in their experiments.
> > >
> > >    - They use **different normalization methods**. For example, LSTNet normalizes raw data row by row using the maximum absolute value, short for max-abs normalization; GraphWaveNet uses Z-score normalization; StemGNN uses Z-score normalization for some datasets and min-max normalization for other datasets.
> > >
> > >    - They use **different data splitting ratios**. For example, LSTNet uses 6:2:2; DCRNN uses 7:2:1; MTGNN and StemGNN use 6:2:2 for some datasets and 7:2:1 for other datasets.
> > >
> > >    - They use **different prediction lengths**. For example, GraphWaveNet uses \{3,6,12\}; STGCN uses \{3,6,9\};
> > > LSTNet uses \{3,6,12,24\}; AGCRN uses \{12\}; MTGNN uses \{1, 12\}; StemGNN uses different prediction lengths for different datasets, such as \{3,12,28\}.
> > >
> > > It is important to note that due to the significant variations in experimental settings among different baselines, we did not directly replicate the results reported in the baseline papers for our paper. In our work,
> > > 1. We first unify the experimental settings to guarantee a fair and more convincing comparison. Specifically, we
> > >    - adopt the min-max normalization and the splitting ratio of 7:2:1 for all datasets,
> > >    - fix the input length 12 and the output length 12 in Table 1, and
> > >    - set the input length 12 and the output length \{3,6,9,12\} for the multi-step forecasting.
> > >
> > >    An exception is made for COVID-19 where we adopt the ratio of 6:2:2 because the number of samples in COVID-19 is too small (355). Please refer to **Lines 240-244** for more details.
> > > 2. We then re-run **all baselines under the above settings on all datasets for both the experiments presented in our paper and those conducted during the rebuttal phase**. Despite the substantial workload and large time consumption, we firmly believe that employing a unified experimental framework and reproducing the results ensures a fairer comparison, ultimately contributing to the advancement of short-term forecasting. We believe our dedicated efforts in conducting these experiments coincide with your rigorous attitude and expectations toward our experimental settings/results.
> > >
> > > Furthermore, due to the different experimental settings, **the results of the baselines reported in our paper may differ from those reported in their original papers**. This discrepancy is particularly prominent for baselines that utilize different normalization methods, such as LSTnet using max-abs normalization as default, StemGNN using Z-score or min-max normalization, and our settings using min-max normalization. This disparity is expected, as data normalized using different methods can have varying scales, while most multivariate time series (MTS) forecasting methods evaluate results on normalized data.

---

> > > > ### Author Response · Authors · 2023-08-19
> > > > **Thanks for your feedback (2/2)**
> > > >
> > > > Thanks for your valuable time and careful review.
> > > >
> > > > **W2**
> > > >
> > > > **The numbers you report for e.g. PatchTST deviate from the published numbers. Can you explain these differences?**
> > > >
> > > > The differences are attributed to the fact that we adopt a different normalization method (**min-max normalization**) in the experiment to that (**standard normalization**) in the PatchTST paper.
> > > > - As we stated in **Lines 241-242**, all datasets are normalized using the **min-max normalization**. Keeping consistent with the setting, we also performed the **min-max normalization** on the four datasets, i.e., Exchange, Weather, ETTh1, and  ETTm1. Accordingly, we rerun PatchTST, DLinear, and our FourierGNN on the normalized data with the same input length of 96. The corresponding results are reported for comparison in **Table 2 in the attached PDF**.
> > > > - In addition, we aim to investigate the long-term forecasting performance of our FourierGNN compared with two SOTA long-term forecasting performances. Under the same experimental settings, the results in Table 2 in the rebuttal PDF verify that our FourierGNN performs worse than the baselines under long-term forecasting settings, which confirms our expectation that our FourierGNN is designed for short-term forecasting and unsuitable for long-term forecasting.
> > > > - Honestly, comparing FourierGNN with PatchTST and DLinear under the settings of the baseline papers would be more appropriate. We reperformed the experiments and rerun our FourierGNN under the same normalization settings of the PatchTST paper. We report the corresponding results in the following table, where we can achieve the same conclusion that FourierGNN does not perform as well as PatchTST and DLinear under large horizons, i.e., the long-term forecasting settings.
> > > >
> > > >
> > > > ETTh1
> > > >
> > > > |  Model| Metric|96 | 192 | 336 | 720 |
> > > > |:--------|:----|:----|:----|:----|:----|
> > > > |PatchTST|MAE| 0.370| 0.413| 0.422| 0.447|
> > > > | |MSE|0.400 | 0.429| 0.440| 0.468|
> > > > |DLinear|MAE| 0.375| 0.405| 0.439| 0.472|
> > > > | |MSE|0.399 | 0.416| 0.443| 0.490|
> > > > |FourierGNN|MAE| 0.433| 0.486| 0.507| 0.564|
> > > > | |MSE| 0.451| 0.553| 0.594| 0.663|
> > > >
> > > > **W3**
> > > >
> > > >
> > > > **1. for the main experiment in tab. 1: what is \tau? 1 ?**
> > > >
> > > > The $\tau$ in Table 1 is 12, and we will mark this in the caption of Table 1.
> > > >
> > > > **2. what does "follow the experimental settings in short-term forecasting baselines, like LSTNet and StemGNN" mean exactly?**
> > > >
> > > > The sentence "follow the experimental settings in short-term forecasting baselines, like LSTNet and StemGNN" in the rebuttal
> > > >   means that we adopt $\tau\in\{3,6,9,12\}$ in our experiments, following the two papers.
> > > >
> > > > **2.1 The LSTNet paper reports two different error measures, RSE and corr, so one cannot compare numbers directly.**
> > > >
> > > > Since the metrics, the datasets, and the experimental settings of LSTNet are different from our paper and some other baseline papers, we re-run the source code of LSTNet according to our experimental settings and datasets. To ensure a comprehensive assessment, we employed three evaluation metrics in MTS forecasting, namely MAE, RMSE, and MAPE. These evaluation metrics are also commonly utilized in MTS forecasting research, as exemplified by the baselines StemGNN and AGCRN.
> > > >
> > > > **2.2 The StemGNN paper reports RMSE for datasets Solar and Electricity, but they report different numbers than you do (their table 2)**
> > > >
> > > > As aforementioned, the different numbers are derived from the different experimental settings between the StemGNN paper and ours. In the StemGNN paper, for Solar and Electricity datasets, the input length is 24 and the prediction length is 3; while in our settings, both the input length and prediction length are 12. Meanwhile, StemGNN uses Z-score normalization, whereas our model employs min-max normalization.
> > > >
> > > > **3. Do any of the numbers in your table 1 coincide with some published numbers in the baseline papers? And if so, could you mark them, say with a star?**
> > > >
> > > > As explained in our responses to the experimental settings, in our work, we adopt the unified experiments settings for all methods for a fair comparison. Since the experimental settings are different in each baseline paper, we re-run all baseline source codes and report the result numbers carefully. Correspondingly, the results of the baselines reported in our paper may differ from those reported in their original papers. We believe we could advance short-term time series forecasting by using such unified settings for fair comparisons.
> > > >
> > > >
> > > > Thanks again for your feedback. Hope we have addressed all your concerns.
> > > >
> > > > Authors

---

### Official Review · Reviewer_G8af · 2023-07-08

**Soundness:** 3 good
**Presentation:** 3 good
**Contribution:** 3 good
**Rating:** 6
**Confidence:** 5

**Summary:**

In this paper, the authors study a problem in GNN-based multivariate time series (MTS) forecasting, i.e. modeling spatial correlations and temporal dependencies in the same time. In particular, the authors do not follow previous works on regarding the input as T graphs and capturing temporal dependencies between graphs by temporal networks, but propose to formulate T graphs as a hypervariate graph(a pure fully-connected graph with NT nodes). Specifically, to deal with the huge node amount NT in a hypervariate graph, the authors propose the FourierGNN which utilize matrix multiplications in the Fourier space of graphs to decrease the quadratic computational complexity to quasi linear one. Empirical results on seven datasets show the effectiveness of this method.

**Strengths:**

1. The authors provide a novel view of graph-based MTS forecasting called hypervariate graph.
2. The authors propose Fourier Graph Operator and FourierGNN for efficient convolution on hypervariate graph.
3. In the authors’ experiments, FourierGNN obtains superior performance.


**Weaknesses:**

1.	The detail of the experiments is not clear. The authors run previous models on different datasets but do not show whether some hyperparameters, such as feature dim, change with the dataset.
2.	The authors claim that FourierGNN is efficient for hypervariate graph, but do not provide other networks’ performance such as GCN on hypervariate graph.
3.	The models compared with Efficiency Analysis are too old.
4.	The authors do not explain the reason for the Green kernel. The Green kernel is a strong assumpition. In line 161, k_{ij} = A_{ij} W and k_{ij} = k_{i-j} results A_{j+k}_{j} equals for any j, which strictly restricts the topology of the hypervariate graph. Besides, there is no visualization of A in each layer to check whether the green kernel assumption is satisfied during training.


**Questions:**

1.	What’s the performance of FourierGNN on longer prediction lengths such as {24, 36, 48, 60} in FEDformer?
2.	What’s the meaning of “the same sparsity pattern” in line 196?
3.	How to determine the value of A_i in equation (7), A_0 is identity and what about A_1, A_2?


**Limitations:**

Please check the weakness

---

> ### Author Rebuttal · Authors · 2023-08-09
>
> We appreciate your positive feedback and valuable suggestions. In the following, we provide a detailed response to address all of your concerns.
>
> **W1.**
>
> Thanks for your suggestion. We provide more clear details to clarify the experimental settings of baselines in Appendix E.2. In the experiments, we 1) followed the parameter configuration recommended by the original authors of each baseline for the datasets used in both the baseline paper and our paper; and 2) tuned the recommended parameter settings on datasets not used in the baseline paper to guarantee that baselines achieve the best results.
>
> Important reproduction details for all baselines are provided in Appendix E.2. For example, the parameter settings of DeepGLO, TAMP-S2GCNets, DCRNN, and STGCN vary with different datasets.
>
> **W2.**
>
> We have conducted a thorough analysis of the time complexity of our proposed Fourier Graph Operator (FGO) in comparison with graph convolution, as presented in **Lines 203-209** of the paper.
>
> To clarify more clearly, we summarize the time complexity and parameter volumes in **Table 5 in the attached PDF**.
>
> Furthermore, to address your concern, we conducted additional experiments on METR-LA and ECG datasets to compare the time complexity of FourierGNN with GCN. In the experiments, we replaced FourierGNN with GCN [1] and performed GCN on the hypervariate graph. We compare FourierGNN with two types of GCN: 1) GCN-N performs GCN on the graph of $N$ variable nodes; 2) GCN-NT performs GCN on the hypervariate graph with $NT$ nodes. Since GCN-N, as a typical GCN, requires a pre-defined graph topology, it can not be conducted on ECG dataset because the dataset has no pre-defined graph topology. For GCN-NT, we input the adjacency matrix with values all one as the graph topology. The corresponding results (average the results of five epoch times) in terms of training time costs on two datasets are reported in the following table:
>
> |    Models    | METR-LA (N=207, T=12)| ECG (N=140, T=12) |
> |:---|:---|:---|
> |  FourierGNN  | 99.76 $\pm$ 2.74 s/epoch | 8.98 $\pm$ 0.31 s/epoch |
> |     GCN-N     | 213.64 $\pm$ 1.21 s/epoch| --- |
> | GCN-NT| 1976.33 $\pm$ 6.24 s/epoch| 384.58 $\pm$ 2.86 s/epoch |
>
> The table reveals a notable efficiency advantage of FourierGNN, even showcasing superior performance compared to GCN with $N$ nodes. We will add this experiment to the appendix of our final version.
>
> [1]. Kipf & Welling, Semi-Supervised Classification with Graph Convolutional Networks, ICLR 2017
>
> **W3.**
>
> Since FourierGNN is a frequency-related GNN-based model, we previously chose the GNN-based baselines for efficiency analysis. We will add two **latest frequency-related** baselines, i.e., Autoformer (2021, NeurIPS) and FEDformer (2022, ICML), in the **Efficiency Analysis of Section 5.3**. The new results are presented in **Table 6 in the attached PDF**.
>
> **W4.**
>
> Note that the hypervariate graph $\mathcal{G}\_t$ is a **fully-connected graph** (i.e., $A=\{1\}^{n\times n}$) as shown in **Definition 1**. Accordingly, we can easily prove the green kernel assumption on the hypervariate graph.
>
> Regarding the hypervariable graph: since the underlying topology structure of the $NT$ nodes is generally not known in advance, we initialize the hypervariate graph with a fully-connected graph and subsequently learn the edge weights (i.e., node correlations or spatiotemporal dependencies) on the graph (more explanation can be seen in **Appendix C**).
>
> In addition, since the hypervariate graph is fully-connected, we have visualized the learned node correlations (edge weights) in each layer on the hypervariate graph in **Figure 9 in Appendix H.1**. Specifically, we have visualized the learned adjacency matrices corresponding to the original spectrum of the input, as well as the outputs of the first, second, and third layers of  FourierGNN in **Figure 9**.
> These results show that our FGO can adaptively and effectively capture important spatiotemporal correlations while removing noises, enabling the learning of a discriminative model.
>
> **Q1.**
>
> To address your concern, we performed additional experiments to compare the performance of FourierGNN and FEDformer on longer prediction lengths \{12,24,36,48,60\} with an input length of 24 on METR-LA dataset. The results are as below:
>
> |  Model| Metric|12 | 24 | 36 | 48 | 60 |
> |:----|:----|:----|:----|:----|:----|:---|
> |FEDformer|MAE| 0.108| 0.120| 0.137| 0.148| 0.163|
> | |RMSE|0.190 | 0.216| 0.231| 0.259| 0.278|
> |FourierGNN|MAE| 0.087| 0.115| 0.140| 0.155| 0.169|
> | |RMSE| 0.169| 0.207| 0.230| 0.265| 0.287|
>
> The results demonstrate that FourierGNN achieves significantly better performance than FEDformer on lower prediction lengths, such as {12, 24}, but underperforms FEDformer on longer prediction lengths \{36,48,60\}. This is reasonable because FEDformer, being a Transformer-based baseline, is effective in capturing long-range temporal dependencies, while FourierGNN similar with GNN-based models is skilful in capturing local/short-term spatiotemporal dependencies.
>
> **Q2.**
>
> The same sparsity pattern means that the positions of non-zero elements in two matrices are identical. The two matrices have the same structure of connections between elements, even though the actual values of the non-zero elements might be different.
>
> **Q3.**
>
> Since the underlying structure of the hypervariate graph is not given, we initialize the hypervariate graph as a fully-connected graph, i.e., $A=\{1\}^{n\times n}$. All adjacency matrices $\{A_i\}_{i=1}^K$ in $K$-order FourierGNN sharing the same sparsity pattern with $A$ are fully-connected. But their edge weights are learned during training our FourierGNN on a specific dataset. This corresponds to learning the spatiotemporal dependencies between nodes.
>
> We will clarify the above in the final version, and hope we have addressed all of your comments.

---

### Author Rebuttal · Authors · 2023-08-09

Dear Reviewers, ACs, and the SAC:

We thank all reviewers for their valuable comments. We response to all comments of reviewers; in particular, after carefully considering ndBv's comments, we realize there might be some potential misunderstandings. We've tried our best to clarify these misunderstandings in the specific rebuttal.

**Short-term vs long-term forecasting presents significant differences**
  + Long-term forecasting focuses on a long historical context to efficiently capture long-range dependencies such as periodic patterns and trends. In contrast, short-term forecasting often involves dealing with rapidly changing and dynamic patterns.
  + In the literature, the benchmark datasets for short-term and long-term forecasting are different. Long-term forecasting involves most datasets containing a few variables, such as Exchange (8 variables), ETTh, ETTm, ILI (7 variables), and Weather (21 variables). In contrast, short-term forecasting datasets contain more variables, for example, the datasets in StemGNN have at least 140 variables except the COVID-19 dataset, and the two datasets in AGCRN have 307 and 170 variables, respectively.
  + In deep networks, SOTA long-term forecasting models are mainly Transformer-based and MLP-based, while SOTA short-term forecasting models are mainly GNN-based models. This is because GNN-based models are more effective at capturing spatial correlations among variables and require sufficient variables to guarantee an appropriately sized graph, while Transformer-based/MLP models are more efficient at processing very long inputs/predictions.

In summary, our FourierGNN aims to learn spatiotemporal dependencies and is designed specifically for short-term forecasting. The experiments are consistent with the literature on short-term forecasting, compared with SOTA GNN-based models, frequency-based models, and short-term models.

**W3**
   1. We follow the experimental settings in short-term forecasting baselines, like LSTNet and StemGNN, and the forecasting horizon $\tau$ in our experiment is set to 3, 6, 9, 12.
   2. Based on the above discussion of the differences between short-term and long-term forecasting, as shown in the literature, it would be generally unfair to compare long-term baselines with short-term baselines, or vice versa. The response to W2 verifies this conclusion that the long-term baselines underperform FourierGNN on short-term forecasting.
   To further address your concerns, we have evaluated FourierGNN's long-term forecasting performance compared with PatchTST and DLinear. The experimental results are shown in **Table 2 in the attached PDF**. From the results, we can find that FourierGNN performs not very well, which is attributed to the fact that FourierGNN focuses on learning the united spatiotemporal dependencies, while long-term forecasting often focuses on long-term periodic patterns and trends.
   3. As discussed above, ETTm2, Exchange, Weather, and ILI are generally used for evaluation in long-term forecasting.

**W4**
1. Actually, in the literature, it is quite prevalent to employ a combination of Fourier transform (FT) and inverse Fourier transform (IFT) operations, including StemGNN, FEDformer, Autoformer, CoST, FiLM [1], and FreDo [2]. They transform time series into the frequency domain via FT, then perform calculations in the frequency domain, and finally conduct IFT to transform back the results to the time domain for subsequent operations.
   - This is because the frequency spectrum in the frequency domain is complex-valued (consisting of a real part and an imaginary part), while data in the time domain is real-valued. It is essential and necessary to perform inverse Fourier transform to transform complex-valued frequency values to real-valued time domain values, other complex-valued outputs can not feed into a traditional (real-valued) neural output layer for real-valued forecasts.
   - According to the convolution theorem, we obtain that the recursive multiplication of FGOs (containing a pair of FT and IFT) in Fourier space is equivalent to multi-order convolutions (see **Proposition 1**). In other words, multi-order convolutions in the time domain can be efficiently conducted by leveraging FT and IFT (see **Eq. 7**), which is the core idea of this work.
   - FT and ITF are linear transforms, which do not add or discard any information.

   As we know, some models, such as BTSF [3] and TF-C [4], incorporate frequency domain and time domain features into enhancing time series representations. Since the frequency features in the frequency domain are complex-valued, to combine them with time domain real-valued features, these models **discard** the phase information and only take the frequency magnitude spectrum as **features**. Since the magnitude spectrum is real-valued, there is no need to perform IFT. They leverage spectrum information as features for **data augmentations**, while in our model, we take advantage of the efficiency of Fourier transform. These are two different paradigms.
2. We further perform an ablation study on METR-LA and ECG datasets to compare FourierGNN with its variants that obtain either the real part or the imaginary part of the frequency spectrum in FourierGNN. In this case, the inverse Fourier transform is removed in FourierGNN's variants. The results are reported in **Table 4 in the attached PDF** which shows the real part is more important than the imaginary part for the performance, and both the real part and the imaginary part are indispensable for FourierGNN.

[1]. FiLM: Frequency improved Legendre Memory Model for Long-term Time Series Forecasting. NeurIPS 2022
[2]. FreDo: Frequency Domain-based Long-Term Time Series Forecasting. CoRR abs/2205.12301 (2022)
[3]. Unsupervised Time-Series Representation Learning with Iterative Bilinear Temporal-Spectral Fusion. ICML 2022.
[4]. Self-Supervised Contrastive Pre-Training For Time Series via Time-Frequency Consistency. NeurIPS 2022

---

### Decision · Program_Chairs · 2023-09-21

**Decision:**

Accept (poster)

**Comment:**

The paper introduces a novel approach for multivariate time series forecasting using Graph Neural Networks (GNNs). In contrast to previous methods, this paper addresses spatial and temporal dependencies by representing data as a hypervariable graph. The proposed FourierGNN employs matrix multiplications in Fourier space to efficiently handle the complexity of hypervariate graphs, achieving higher accuracy and efficiency on multiple datasets.

Reviewers raise concerns about the experimental setting as well as the reproduction of SOTA baselines. However, the value of the contributions outweighs these concerns. Acceptance is recommended, and it is encouraged to integrate the reviewers' feedback during the preparation of the final version of the paper.